# Suppression of TGF-β/SMAD signaling by an inner nuclear membrane phosphatase complex

Zhe Ji [1,4], Wing-Yan Skyla Siu [1,4], Maria Emilia Dueñas [2,3], Leonie Müller [2], Matthias Trost [2] & Pedro Carvalho [1] ✉

Cytokines of the TGF-β superfamily control essential cell fate decisions via receptor regulated SMAD (R-SMAD) transcription factors. Ligand-induced R-SMAD phosphorylation in the cytosol triggers their activation and nuclear accumulation. We determine how R-SMADs are inactivated by dephosphorylation in the cell nucleus to counteract signaling by TGF-β superfamily ligands. We show that R-SMAD dephosphorylation is mediated by an inner nuclear membrane associated complex containing the scaffold protein MAN1 and the CTDNEP1-NEP1R1 phosphatase. Structural prediction, domain mapping and mutagenesis reveals that MAN1 binds independently to the CTDNEP1-NEP1R1 phosphatase and R-SMADs to promote their inactivation by dephosphorylation. Disruption of this complex causes nuclear accumulation of R-SMADs and aberrant signaling, even in the absence of TGF-β ligands. These findings establish CTDNEP1-NEP1R1 as the R-SMAD phosphatase, reveal the mechanistic basis for TGF-β signaling inactivation and highlight how this process is disrupted by disease-associated MAN1 mutations.

SMADs constitute a family of transcription factors in animals, essential during embryonic development and adulthood. As key effectors of the transforming growth factor β (TGF-β) superfamily of cytokines, such as TGF-β and Bone Morphogenic Proteins (BMPs), SMADs control a wide range of biological processes like cell differentiation, proliferation, death, adhesion, and migration[1–4]. With such broad and critical roles, it is not surprising that mutations in SMAD proteins have been associated with many diseases, including cancer, fibrosis, and developmental abnormalities[5].

SMAD-dependent gene expression is activated upon the binding of a TGF-β superfamily cytokine to a cognate receptor at the surface of a target cell. Ligand binding stimulates the receptor kinase activity and the subsequent phosphorylation of a receptor-regulated SMAD, or R-SMAD, at conserved serine residues in a C-terminal SXS motif. Once phosphorylated, R-SMADs bind to SMAD4, also known as the common SMAD or Co-SMAD. These SMAD complexes accumulate in the nucleus, where they act as transcription factors to regulate gene expression[2].

The diversity of transcriptional outputs triggered by TGF-β superfamily cytokines depends largely on the specificity of ligand-receptor interactions and the existence of multiple R-SMADs[4,6]. For example, while BMP ligands predominantly activate the R-SMADs 1, 5, and 8, whereas TGF-β and activin ligands primarily activate SMAD2 and SMAD3. Thus, despite a common activation mechanism, the diversity of SMAD complexes leads to a wide range of transcriptional outputs.

Activation of R-SMADs by phosphorylation in the cytoplasm is counteracted by dephosphorylation of the C-terminal SXS motif. Based on photobleaching and mathematical modeling experiments, R-SMAD inactivation by dephosphorylation depends on a nuclear localized phosphatase[7]. Earlier studies suggested that the protein phosphatase Mg2+/Mn2+ dependent 1A (PPM1A) dephosphorylates R-SMADs to terminate signaling by TGF-β superfamily cytokines[8,9].

[1]Sir William Dunn School of Pathology, University of Oxford, Oxford, UK. [2]Biosciences Institute, Newcastle University, Newcastle upon Tyne, UK. [3]Present address: The Kids Research Institute Australia, Perth Children's Hospital, Nedlands, Australia. [4]These authors contributed equally: Zhe Ji, Wing-Yan Skyla Siu. ✉e-mail: pedro.carvalho@path.ox.ac.uk

However, PPM1A appears to affect R-SMADs indirectly by facilitating their nuclear export[10]. Consistent with an indirect role, the PPM1A knockout (KO) mouse is viable and does not show the developmental defects typical of deregulated TGF-β/SMAD signaling[11–13].

Genetic studies in mice[14–17], frogs[18,19] and flies[20–22]. Implicated the serine/threonine protein phosphatase CTDNEP1 (C-terminal domain nuclear envelope phosphatase, also known as Dullard) in TGF-β signaling. In all cases, loss-of-function mutations in CTDNEP1 resulted in increased TGF-β/SMAD signaling, highlighting its importance as a negative regulator of this pathway. While the mechanism by which CTDNEP1 inhibits TGF-β signaling remains unknown, studies in flies have shown that CTDNEP1 interacts with R-SMADs and may facilitate their dephosphorylation[20].

In humans, mutations in CTDNEP1 are frequently observed in aggressive forms of medulloblastoma that display amplification of the C-MYC oncogene[21–23]. Interestingly, TGF-β signaling is also elevated in these aggressive tumors[23].

In cells, CTDNEP1 associates with the endoplasmic reticulum (ER) membrane, including the nuclear envelope, via an amphipathic helix[24–26] and has a well-established and evolutionarily conserved role in dephosphorylating Lipin, a phosphatidic acid hydrolase, thereby contributing to lipid homeostasis[24,27,28]. Consistent with its role in Lipin regulation, mutations in CTDNEP1 result in excessive ER membrane proliferation in various cell types[28,29]. Recently, CTDNEP1 was also shown to control the stability of SUN2, an inner nuclear membrane (INM) protein important for nuclear organization by transducing cytoskeletal forces into the nucleus[25,30]. These functions of CTDNEP1 also depend on NEP1R1 (nuclear envelope phosphatase 1 regulatory subunit 1), a small ER/NE membrane protein that stabilizes CTDNEP1[26,31]. Whether NEP1R1 also functions with CTDNEP1 to regulate TGF-β/SMAD signaling is unknown.

Here, we showed that CTDNEP1-NEP1R1 phosphatase complex dephosphorylate R-SMADs at the C-terminal SXS motif to inactivate signaling by TGF-β ligands. This dephosphorylation occurs in the nucleus and depends on the binding of CTDNEP1-NEP1R1 to the INM protein MAN1, which acts as an R-SMAD-specific adapter. Disruption of CTDNEP1-NEP1R1-MAN1 complex results in spontaneous nuclear accumulation of R-SMADs and aberrant signaling. These findings provide insight into the mechanism of R-SMAD regulation and explain the molecular basis of diseases and developmental defects caused by unrestrained TGF-β signaling.

## Results

### The CTDNEP1-NEP1R1 phosphatase interacts with the inner nuclear membrane protein MAN1

As a first step in studying the function of CTDNEP1, we searched for its binding partners. CTDNEP1 was fused to a triple FLAG epitope tag (CTDNEP1-FLAG) and expressed in HeLa cells. Like endogenous CTDNEP1[25], CTDNEP1-FLAG localized throughout the ER and the nuclear envelope (Fig. S1A). Lysates from cells expressing CTDNEP1-FLAG were prepared in a buffer containing 1% DMNG, subjected to immunoprecipitation with anti-FLAG beads, and co-precipitated proteins were analyzed by mass spectrometry (Fig. 1A, Source data_proteomics). Among the most enriched proteins in the CTDNEP1-FLAG precipitates was its regulatory subunit NEP1R1, suggesting that the fusion protein was functional. The CTDNEP1-FLAG precipitates were also highly enriched in MAN1 (also known as LEMD3), an understudied INM resident protein with links to TGFβ/BMP signaling[32–35] and mutated in bone disorders characterized by excessive TGFβ/BMP signaling[36–39].

The interaction between CTDNEP1-FLAG and MAN1 was confirmed using immunoprecipitation followed by blotting with anti-MAN1 antibodies (Fig. 1B). A mutant that abolishes CTDNEP1 catalytic activity (CTDNEP1^{D67E, D69T}) also co-precipitated with MAN1 indicating that the interaction was independent of CTDNEP1 phosphatase activity

(Fig. 1B). Similar immunoprecipitations were performed with CTDNEP1's regulatory subunit, NEP1R1, expressed as a fusion to HA epitope tag (NEP1R1-HA). NEP1R1-HA localized throughout the ER and nuclear envelope, as expected (Fig. S1B). As CTDNEP1, NEP1R1-HA also co-precipitated endogenous MAN1 (Fig. 1C). Conversely, immunoprecipitation of V5-tagged MAN1 (V5-MAN1), which localized to the nuclear rim as was endogenous MAN1 (Fig. S1C), interacted with both CTDNEP1-FLAG (Fig. S1D) and NEP1R1-HA (Fig. S1E). Importantly, interactions between the CTDNEP1-NEP1R1 phosphatase complex and MAN1 appeared specific since SUN1 and other abundant INM proteins were not present in the precipitates, as assayed by immunoblot (Figs. 1B, C, S1D, E) and mass spectrometry (Source data_proteomics). Altogether, these data indicate that MAN1 is a binding partner of the CTDNEP1-NEP1R1 phosphatase complex, likely at the INM, where all three proteins were shown to localize.

### R-SMADs interact with the CTDNEP1-NEP1R1 phosphatase complex

To further explore the role of MAN1, we also analyzed its interactors using V5-MAN1 co-immunoprecipitation followed by mass spectrometry, as described for CTDNEP1. In agreement with our data above, endogenous CTDNEP1 and NEP1R1 were among the most enriched proteins in the V5-MAN1 precipitates (Fig. 1D, Source data_proteomics). Other prominent interactors of MAN1 were the R-SMADs 1,2,3, and 5, while no interaction was detected with the co-SMAD SMAD4, as previously observed[32,33,40].

These results raised the possibility that the CTDNEP1-NEP1R1 phosphatase complex, MAN1, and R-SMADs were all part of the same protein complex. Consistent with this idea, we observed that CTDNEP1-FLAG (Figs. 1E, S1F) and NEP1R1-HA (Figs. 1F, S1G) co-precipitated both V5-MAN1 and the R-SMADs SMAD1 and 2. In contrast, the Co-SMAD SMAD4 was not detected in the precipitates. Interestingly, CTDNEP1-NEP1R1 interactions with SMAD2 (Fig. 1E, F) and SMAD1 (Fig. S1F, G) were observed under basal conditions and were only slightly increased by stimulating SMAD2 and SMAD1 signaling with TGF-β and BMP cytokines, respectively. Together, these data indicate that R-SMADs interact with the CTDNEP1-NEP1R1 phosphatase complex.

### CTDNEP1, NEP1R1, and MAN1 are required for R-SMAD dephosphorylation

R-SMADs are activated by phosphorylation on a conserved SXS motif at their extreme C-termini, resulting in their nuclear accumulation[1]. How R-SMADs are inactivated by dephosphorylation remains controversial and mysterious[13,41]. Our finding that R-SMADs form a complex with CTDNEP1-NEP1R1 phosphatase and MAN1 prompted us to test whether these proteins were required for R-SMAD dephosphorylation. A pool of phosphorylated R-SMADs was generated by stimulating cells with TGF-β or BMP, and the kinetics of R-SMAD dephosphorylation and R-SMAD localization were analyzed after washing out the activating ligands. Inhibitors of the kinase activity of TGF-β (SB431542) or BMP (LDN 193189) cell surface receptors were also added to prevent R-SMAD re-phosphorylation (Fig. 2A), as previously described[8]. In parental HeLa, phospho-SMAD2 had a half-life of less than 60 min. Similarly, deletion of the phosphatase PPM1A, previously implicated in SMAD dephosphorylation[8], did not affect the kinetics of SMAD2 dephosphorylation (Figs. 2B, S2A). In contrast, loss of MAN1, CTDNEP1, or NEP1R1 strongly inhibited SMAD2 dephosphorylation without affecting the overall levels of SMAD2 protein (Figs. 2B, S2A). Depletion of MAN1, CTDNEP1, or NEP1R1 also resulted in delayed SMAD2 dephosphorylation in U2OS cells (Fig. S2B). Consistent with delayed R-SMAD dephosphorylation, MAN1, CTDNEP1, and NEP1R1 KO cells showed persistent SMAD2 nuclear accumulation 7 h after TGF-β inhibition, while in control cells, SMAD2 nuclear accumulation largely dissipated within 1 h after TGF-β inhibition (Fig. S2C, D).

 

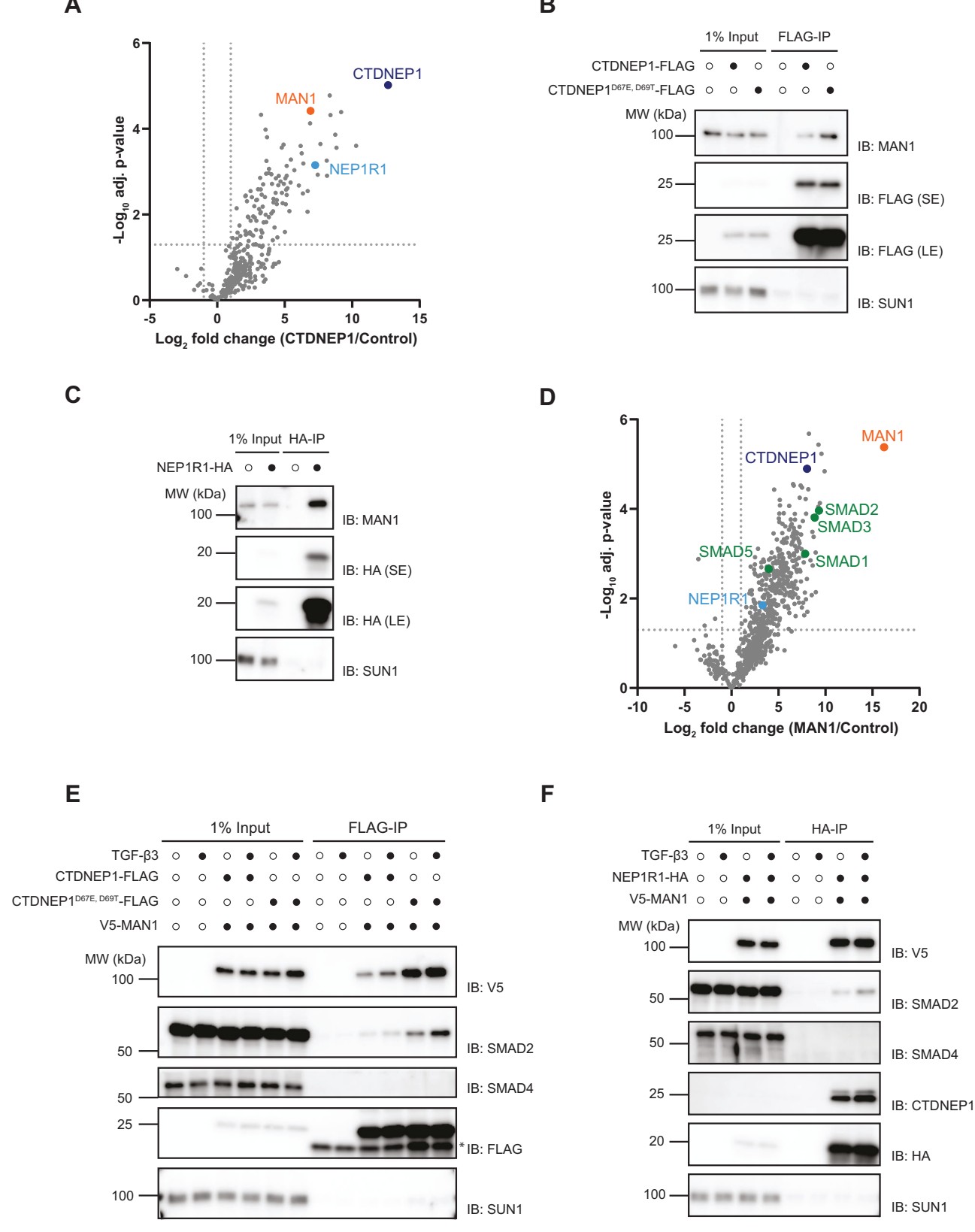

Moreover, MAN1, CTDNEP1, and NEP1R1 KO cells also displayed slowed kinetics of SMAD1/5/8 dephosphorylation after activation with BMP (Fig. 2C). Altogether, these data indicate that upon stimulation with TGF-β superfamily ligands, R-SMAD dephosphorylation requires MAN1, CTDNEP1, and NEP1R1.

Using the assay described above, we tested whether CTDNEP1 phosphatase activity was necessary for R-SMAD dephosphorylation. We observed that re-expression of wild-type CTDNEP1 in CTDNEP1 KO cells restored normal kinetics of SMAD2 dephosphorylation. In contrast, expression of catalytically inactive CTDNEP1$^{D67E, D69T}$ failed to

**Fig. 1 | An inner nuclear membrane complex composed of CTDNEP1-NEP1R1-MAN1 interacts with R-SMADs. A** Proteins co-precipitating with CTDNEP1-FLAG as detected by mass spectrometry. The x-axis shows the log2 fold change of CTDNEP1-FLAG versus untagged control cell line; the y-axis shows the −log10 p-value estimated by the Significance B analysis[71]. CTDNEP1 partner NEP1R1 is labeled in light blue and MAN1 in orange. A two-sided t-test was performed, and the results plotted as a volcano plot in R (significance cut-off: −1 and 1 log2 fold change, −log10 adj.p-value = 1.3). **B**, **C** Immunoprecipitation of FLAG-tagged wild-type CTDNEP1 or phosphatase dead CTDNEP1$^{D67E, D69T}$ (**B**) or NEP1R1-HA (**C**) from detergent solubilized extracts of HeLa cells. Eluted proteins were analyzed by SDS-PAGE followed by immunoblotting with the indicated antibodies. Short and long exposures are shown and labeled SE and LE, respectively. **D** Proteins co-

precipitating with V5-MAN1 as detected by mass spectrometry. The x-axis shows the log2 fold change of V5-MAN1 versus untagged control cell line; the y-axis shows the −log10 p-value estimated by the Significance B analysis[71]. CTDNEP1 and NEP1R1 are labeled in dark and light blue, respectively, and the R-SMADs are labeled in green. A two-sided t-test was performed, and the results plotted as a volcano plot in R (significance cut-off: −1 and 1 log2 fold change, −log10 adj.p-value = 1.3). **E** and **F** Immunoprecipitation of CTDNEP1 or CTDNEP1$^{D67E, D69T}$-FLAG (**E**) or NEP1R1-HA (**F**) from detergent solubilized extracts of HeLa cells co-expressing V5-MAN1. Immunoprecipitations were performed in the absence or upon 1 hr treatment with 20 ng/ml of TGF-β3. Eluted proteins were analyzed by SDS-PAGE followed by immunoblotting with the indicated antibodies. The asterisk (*) indicates the light chain of the antibody used for immunoprecipitation.

promote SMAD2 dephosphorylation (Fig. 2D). Importantly, CTDNEP1$^{D67E, D69T}$ is expressed to normal levels and appears to interact normally with its partners, including SMAD2 (Fig. 1E). Thus, CTDNEP1 phosphatase activity is essential for R-SMAD dephosphorylation. Moreover, overexpression of the phosphatase PPM1A failed to compensate for the absence of CTDNEP1, indicating that CTDNEP1 has a specific role in R-SMAD dephosphorylation (Fig. 2D).

Next, we asked if CTDNEP1 was able to directly dephosphorylate R-SMADs. To this end, soluble derivatives of CTDNEP1 and CTDNEP1$^{D67E, D69T}$ (lacking the N-terminal 45 amino acids encoding an amphipathic helix) were expressed in *E. coli* as His-SUMO tag fusion proteins and affinity purified (Fig. S3A). Purified phospho-SMAD2 immobilized on beads was incubated with recombinant CTDNEP1 or CTDNEP1$^{D67E, D69T}$. While SMAD2 was dephosphorylated by CTDNEP1 in a concentration-dependent manner, CTDNEP1$^{D67E, D69T}$ was unable to do so even when present at high concentrations (Fig. 2E). Collectively, these experiments indicate that R-SMAD dephosphorylation requires INM protein MAN1 and the CTDNEP1-NEP1R1 phosphatase.

## MAN1 is an R-SMAD adapter for the CTDNEP1-NEP1R1 phosphatase

Our data are consistent with a model in which the CTDNEP1-NEP1R1-MAN1 complex functions as the elusive R-SMAD phosphatase. To explore this possibility, we investigated how interactions among CTDNEP1, NEP1R1, and MAN1 contributed to R-SMAD dephosphorylation.

Given that MAN1 was shown to bind R-SMADs directly[42], a simple possibility was that MAN1 bridged the interaction between the R-SMADs and the CTDNEP1-NEP1R1 phosphatase complex. Indeed, in MAN1 KO cells, both CTDNEP1-FLAG (Fig. 3A) and NEP1R1-HA (Fig. 3B) failed to interact with the SMAD1 and SMAD2, even upon TGF-β stimulation that leads to their nuclear accumulation and activation of SMAD signaling. Importantly, loss of MAN1 did not affect the localization of CTDNEP1 (Fig. 3C) or its interaction with NEP1R1 (Fig. 3B). To test whether loss of MAN1 had a general effect on CTDNEP1-NEP1R1 activity we analyzed the levels of Lipin1 and SUN2, two proteins with reduced steady state levels in the absence of CTDNEP1-NEP1R1 phosphatase[25,28,30]. In contrast with CTDNEP1 and NEP1R1 KO cells, MAN1 KO HeLa cells display normal levels of both Lipin1 and SUN2, indicating that MAN1 is not a general regulator of CTDNEP1-NEP1R1 activities and selectively regulates R-SMADs dephosphorylation (Fig. S3B). Together, these data suggest that MAN1 acts as an INM scaffold, specifically facilitating the interaction between R-SMADs and the CTDNEP1-NEP1R1 phosphatase complex.

A prediction of this model was that MAN1 would interact independently with R-SMADs and the CTDNEP1-NEP1R1 phosphatase complex. We explored this possibility by the interactions between MAN1 and the CTDNEP1-NEP1R1 phosphatase complex using immunoprecipitation. Interestingly, regulatory and catalytic subunits showed distinct dependencies in their binding to MAN1. While the interaction between CTDNEP1 with MAN1 required NEP1R1 (Fig. 4A), in CTDNEP1 KO cells, NEP1R1 and MAN1 still interacted, albeit less

efficiently (Fig. 4B). This suggested a central role of NEP1R1 in the recruitment of the CTDNEP1-NEP1R1 phosphatase complex to MAN1. In agreement with these findings, structural analysis using Alphafold multimer predicted an interaction between NEP1R1 and MAN1 through their membrane regions (Fig. S4A, B)[43,44]. To gain further insight into the NEP1R1-MAN1 binding mechanism, we tested these predictions by generating chimeric proteins in which either one or both of the MAN1 transmembrane segments were replaced by the ones of its paralogue LEMD2 (Fig. 4C)[45]. Despite a similar domain organization and inner nuclear membrane localization, LEMD2 functions in maintaining nuclear envelope integrity by facilitating its reassembly during mitosis and repair upon damage[46–48]. Remarkably, both MAN1$^{LEMD2TM1}$ and MAN1$^{LEMD2TM1+2}$ failed to interact with NEP1R1 (Fig. 4D). Like MAN1, the MAN1$^{LEMD2TM}$ mutants localized to the nuclear envelope (Fig. S4C) and interacted normally with R-SMADs (Fig. 4E). Therefore, MAN1 interaction with the CTDNEP1-NEP1R1 phosphatase complex depends on the binding between NEP1R1 and MAN1 transmembrane regions.

Next, we examined the contribution of the MAN1 C-terminal region for the interactions with the phosphatase complex and R-SMADs. MAN1 C-terminus encompasses the UHM domain previously shown to bind to R-SMADs[32,33,42]. We observed that MAN1ΔC, a MAN1 truncation lacking its last 155 amino acids, including the UHM domain, still localized to the INM (Fig. S4C) but failed to interact with R-SMADs (Fig. 4E), as expected. On the other hand, MAN1ΔC efficiently interacted with the CTDNEP1-NEP1R1 phosphatase complex (Fig. 4D, E). Thus, MAN1 functions as a scaffold using distinct domains to interact with R-SMADs and CTDNEP1-NEP1R1 phosphatase complex.

Finally, we assessed the importance of these MAN1 domains for SMAD2 dephosphorylation. The SMAD2 dephosphorylation defect in MAN1 KO cells was reversed by expression of wild-type MAN1 but not by expression of MAN1$^{LEMD2TM1+2}$ or MAN1ΔC, impaired in CTDNEP1-NEP1R1 and R-SMAD binding, respectively (Fig. 4F). Since these MAN1 derivatives display similar levels and localization (Fig. S4C, D), we concluded that MAN1-dependent R-SMAD dephosphorylation requires its binding both to the CTDNEP1-NEP1R1 and R-SMADs. Together, these results indicate that MAN1 functions as an R-SMAD-specific adapter for the CTDNEP1-NEP1R1 phosphatase at the INM.

## CTDNEP1, NEP1R1, and MAN1 suppress inappropriate SMAD signaling

Activation of R-SMADs with TGF-β family cytokines leads to their phosphorylation and nuclear accumulation[1,49]. Intriguingly, depletion of CTDNEP1, NEP1R1, or MAN1 triggered nuclear accumulation of SMAD2 even in the absence of exogenous TGF-β ligands (Fig. 5A, B). This effect appeared specific as SMAD2 distribution was unaffected by the depletion of the phosphatase PPM1A (Fig. 5A, B). The normal SMAD2 distribution was restored upon re-expression of the wild-type in the corresponding KO cells (Fig. S5A, B). On the other hand, in CTDNEP1 KO cells, SMAD2 nuclear accumulation was not reversed by the expression of phosphatase dead CTDNEP1 or by active PPM1A, highlighting the importance of CTDNEP1 catalytic activity in regulating SMAD2 distribution (Fig. S5A, B). Given that R-SMADs nuclear

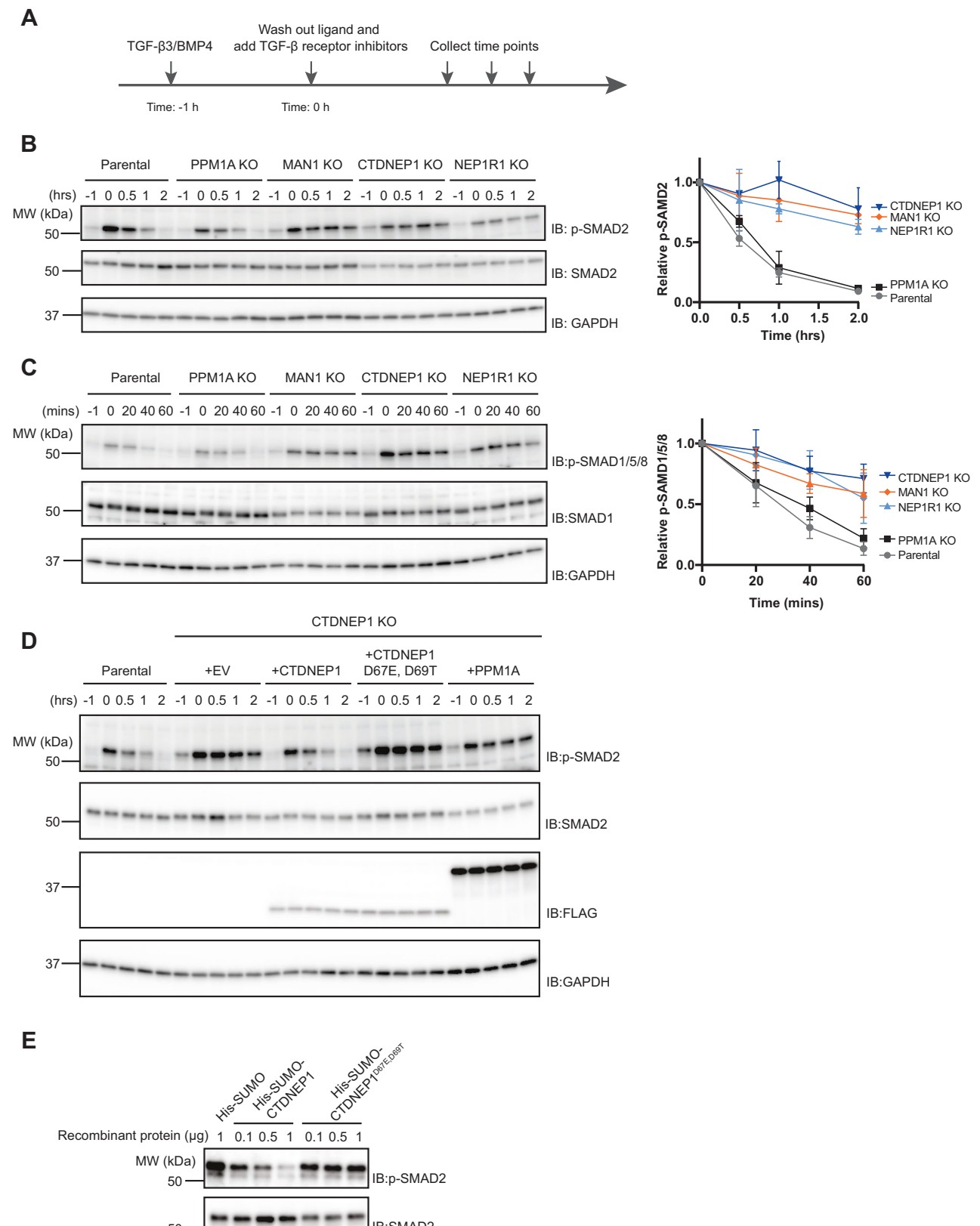

localization requires their prior phosphorylation at the C-terminal SXS motif by receptor-activated kinases, we tested if, in the absence of exogenous TGF-β ligands, CTDNEP1, NEP1R1, or MAN1 KO cells showed increased SMAD2 phosphorylation. Indeed, under these basal conditions, the KO cells showed increased levels of phosphorylated SMAD2, which was undetectable in control cells (Fig. 5C).

Increased levels of phosphorylated SMAD2 in the KO cells could result from autocrine signaling due to increased production of TGF-ß ligands, as previously described[50,51]. Consistently, MAN1, CTDNEP1, and NEP1R1 KO cells expressed higher levels of TGF-β1 and 2 ligands, as detected by RT-qPCR (Fig. S5D). The addition of a neutralizing antibody 1D11, which specifically binds to and inhibits TGF-β ligands[52], to

**Fig. 2 | R-SMAD dephosphorylation requires MAN1 and the CTDNEP1-NEP1R1 phosphatase. A** Scheme of the experimental outline to monitor the kinetics of R-SMAD dephosphorylation. **B** Time course analysis of SMAD2 dephosphorylation upon TGF-β3 stimulation in parental, PPM1A, MAN1, CTDNEP1, and NEP1R1 KO HeLa cells. Cell extracts were analyzed by SDS-PAGE and immunoblotting with the indicated antibodies. The graph (right) shows the average of three experiments; error bars represent the standard deviation. **C** Time course analysis of SMAD1/5/8 dephosphorylation upon BMP-4 stimulation in parental, PPM1A, MAN1, CTDNEP1, and NEP1R1 KO HeLa cells. Cell extracts were analyzed by SDS-PAGE and immunoblotting with the indicated antibodies. The graph (right) shows the average of

four experiments; error bars represent the standard deviation. **D** Time course analysis of SMAD2 dephosphorylation upon TGF-β3 stimulation in parental and CTDNEP1 KO HeLa cells expressing the indicated proteins. Cell extracts were analyzed by SDS-PAGE and immunoblotting with the indicated antibodies. Cells transduced with an empty vector (EV) were used as a control. **E** Analysis of SMAD2 dephosphorylation by recombinant wild-type CTDNEP1 and phosphatase dead CTDNEP1$^{D67E, D69T}$ expressed as fusion proteins to a His-SUMO tag. Purified His-SUMO was also used as a negative control. Note that soluble versions of CTDNEP1 and of CTDNEP1$^{D67E,D69T}$ were generated by deleting the N-terminal amphipathic helix of CTDNEP1 corresponding to amino acids 1–45.

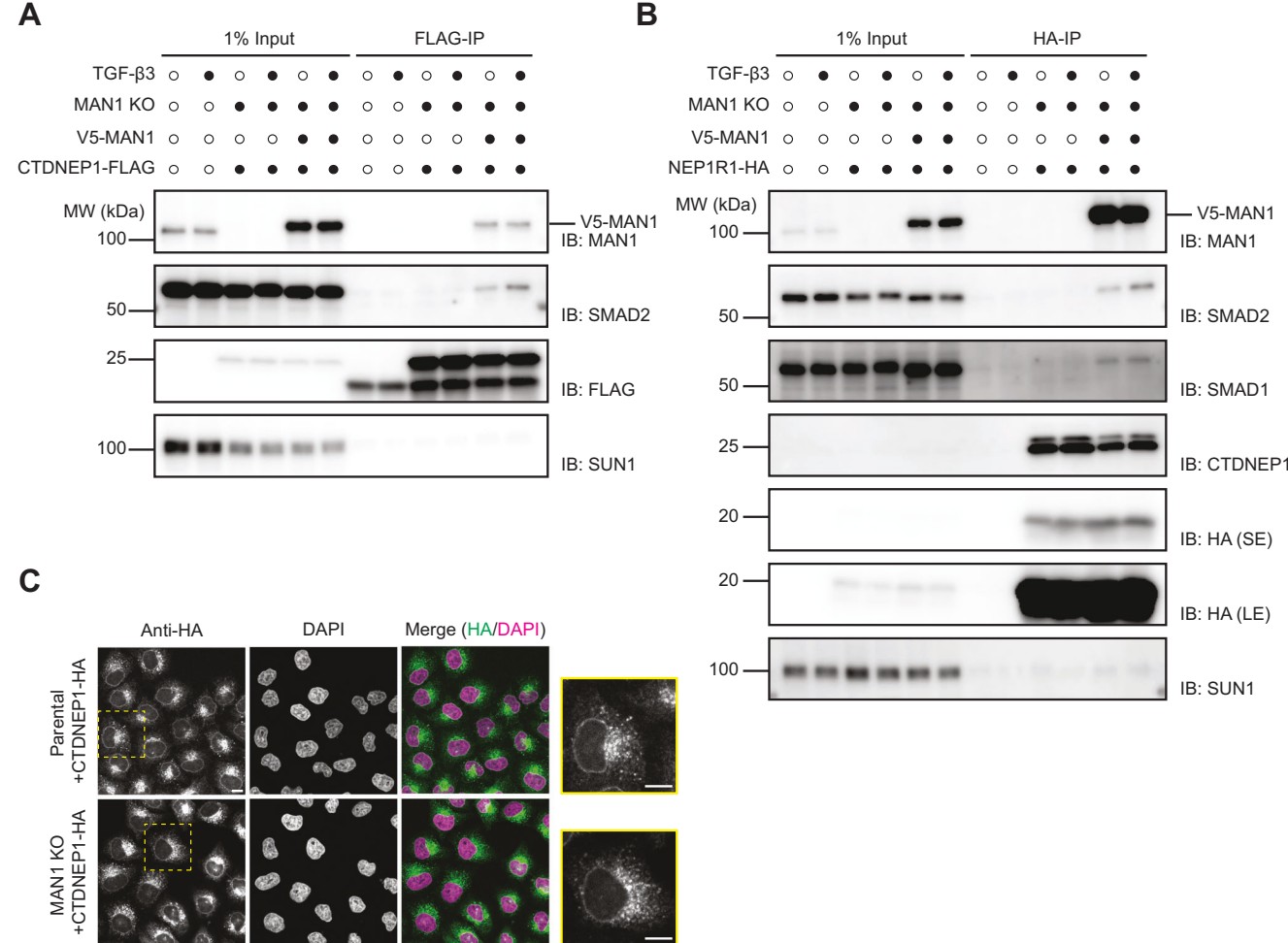

**Fig. 3 | MAN1 is required for the interaction between CTDNEP1-NEP1R1 phosphatase and R-SMADs. A** Immunoprecipitation of FLAG-tagged CTDNEP1 from detergent solubilized extracts of MAN1 KO HeLa cells transduced with an empty vector or a construct encoding V5-MAN1. Immunoprecipitations were performed in the absence or upon a 1 hr treatment with 20 ng/mL of TGF-β3, as indicated. Eluted proteins were analyzed by SDS-PAGE followed by immunoblotting with the indicated antibodies. **B** Immunoprecipitation of HA-tagged NEP1R1 from detergent solubilized extracts of MAN1 KO HeLa cells transduced with an empty vector or a

construct encoding V5-MAN1. Immunoprecipitations were performed in the absence or upon a 1 h treatment with 20 ng/ml of TGF-β3, as indicated. Eluted proteins were analyzed by SDS-PAGE followed by immunoblotting with the indicated antibodies. Short and long exposures are shown and labeled SE and LE, respectively. **C** Localization of CTDNEP1-HA in parental and MAN1 KO HeLa cells analyzed by immunofluorescence. CTDNEP1-HA was detected with anti-HA antibodies and DNA was labeled with 4′,6- diamidino- 2- phenylindole (DAPI). Dotted yellow square indicates the zoomed-in cell shown on the far right. Scale bar: 10 µM.

the cell media reduced the levels of phosphorylated SMAD2 in CTDNEP1 KO cells, further suggesting they have aberrant autocrine TGF-β signaling (Figs. 5D, S5C). However, the neutralizing antibody, even when added in large excess, could not efficiently suppress the SMAD2 phosphorylation in CTDNEP1 KO cells, as observed with the TGF-β receptor kinase inhibitors SB431542 (Fig. 5D). These data suggested that the TGF-β receptor has some basal kinase activity, normally counteracted by the phosphatase activity of CTDNEP1.

Next, we asked if the accumulation of phosphorylated SMAD2 in the KO cells was sufficient to trigger the expression of several SMAD target genes (Fig. 5E), including the cell cycle inhibitors p15, p21 (Fig. 5E, F). Importantly, the accumulation of cell cycle inhibitors was functionally relevant as the KO cells displayed growth inhibition in the absence of TGF-β ligands (Fig. 5G). Therefore, besides the essential role in R-SMAD inactivation upon TGF-β ligand stimuli, the CTDNEP1-NEP1R1-MAN1 complex also has a critical and constitutive function in

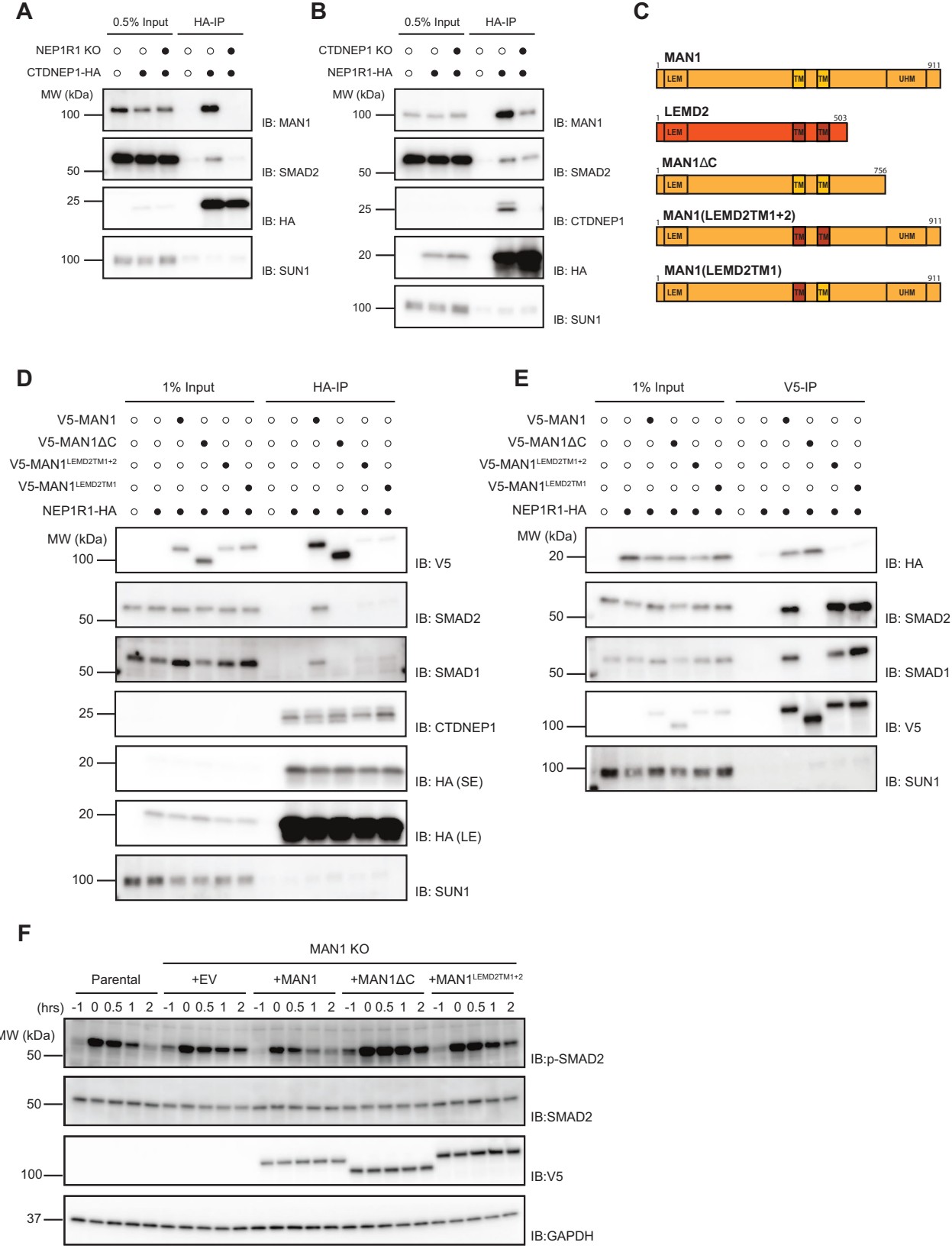

suppressing aberrant SMAD signaling, due to both autocrine and basal activation of the TGF-β receptor.

## Discussion

Accurate signaling via TGF-β superfamily cytokines depends on tight regulation of SMAD activity. While the mechanisms of SMAD activation

by phosphorylation have been extensively characterized, much less is known about their inactivation. Here, we determined the mechanism of R-SMAD inactivation by dephosphorylation and identified CTDNEP1-NEP1R1 as the long-sought R-SMAD phosphatase.

We showed that the dephosphorylation of R-SMAD requires their interaction with an INM complex composed of MAN1 and the

**Fig. 4 | MAN1 binds CTDNEP1-NEP1R1 phosphatase and R-SMADs independently. A** Immunoprecipitation of HA-tagged CTDNEP1 from detergent solubilized extracts in parental or NEP1R1 KO HeLa cells. Eluted proteins were analyzed by SDS-PAGE followed by immunoblotting with the indicated antibodies. **B** Immunoprecipitation of HA-tagged NEP1R1 from detergent solubilized extracts in parental or CTDNEP1 KO HeLa cells. Eluted proteins were analyzed by SDS-PAGE followed by immunoblotting with the indicated antibodies. **C** Schematic representation of MAN1 and related proteins. **D** Immunoprecipitation of HA-tagged NEP1R1 from detergent solubilized extracts of MAN1 KO HeLa cells transduced with an empty vector or the indicated V5-tagged MAN1 derivatives. Eluted proteins were

analyzed by SDS-PAGE followed by immunoblotting with the indicated antibodies. Short and long exposures are shown and labeled SE and LE, respectively. **E** Immunoprecipitation of the indicated V5-tagged MAN1 derivatives from detergent solubilized extracts of MAN1 KO HeLa cells co-expressing HA-tagged NEP1R1. Eluted proteins were analyzed by SDS-PAGE followed by immunoblotting with the indicated antibodies. **F** Time course analysis of SMAD2 dephosphorylation upon TGF-β3 stimulation in parental and MAN1 KO HeLa cells expressing the indicated proteins. Cell extracts were analyzed by SDS-PAGE and immunoblotting with the indicated antibodies. Cells transduced with an empty vector (EV) were used as a control.

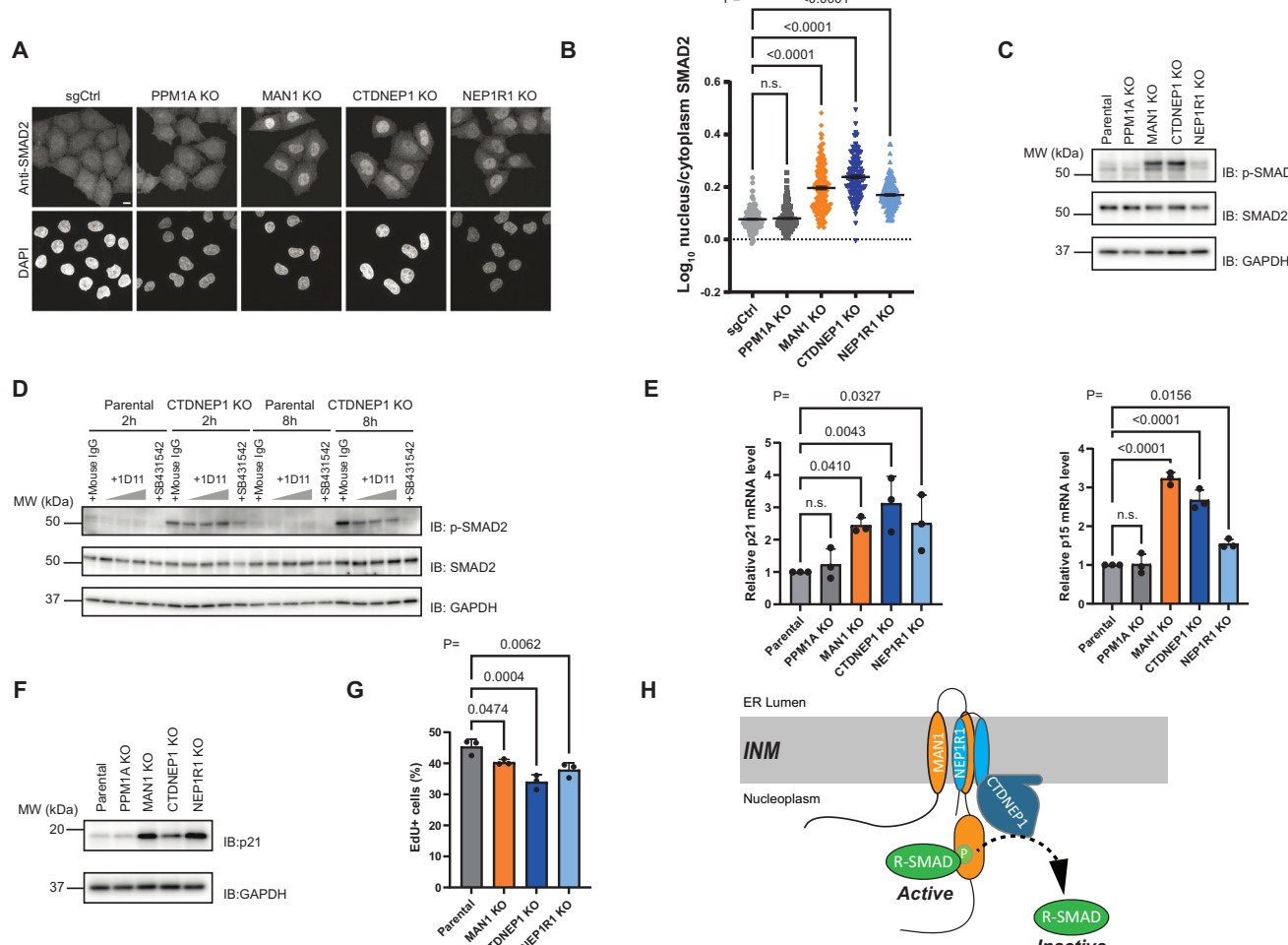

**Fig. 5 | The CTDNEP1-NEP1R1 phosphatase and MAN1 suppress inappropriate SMAD signaling. A** Localization of endogenous SMAD2 in HeLa parental cells or lacking the indicated genes analyzed by immunofluorescence. DNA was labeled with 4′,6- diamidino- 2- phenylindole (DAPI). Scale bar: 10 μM. **B** Quantification of nuclear accumulation of SMAD2 from imaging experiments, as shown in (**A**), n = 3 independent experiments, p-values were indicated in the graph. One-way ANOVA (multiple comparison) was performed, and data are presented as mean values ± SD. **C** Levels of endogenous pSMAD2 and SMAD2 in HeLa parental cells or lacking the indicated genes. Cell lysates were analyzed by SDS-PAGE followed by immunoblotting with the indicated antibodies. GAPDH was used as a loading control. **D** Levels of endogenous pSMAD2 and SMAD2 in HeLa parental cells or CTDNP1 KO cells with indicated treatments. The 1D11 antibody against TGF-β cytokine was used at 30, 150, and 300 μg/mL; mouse IgG antibody was used at 300 μg/mL; the TGF-β receptor specific inhibitor SB431542 was used at 10 μM. Cell lysates were analyzed

by SDS-PAGE followed by immunoblotting with the indicated antibodies. GAPDH was used as a loading control. **E** levels of p21, p15 transcripts in HeLa parental cells or lacking the indicated genes analyzed by RT-qPCR. n = 3 independent experiments, each with 3 technical replicates. p-values were indicated in the graph. One-way ANOVA (multiple comparison) was performed, and data are presented as mean values ± SD. **F** levels of endogenous p21 in HeLa parental cells or lacking the indicated genes. Cell lysates were analyzed by SDS-PAGE followed by immunoblotting with the indicated antibodies. GAPDH was used as a loading control. **G** Quantification of S phase cells assessed based on EdU incorporation and DAPI staining. At least 30000 cells were analyzed for each condition. n = 3 independent experiments, p-values were indicated in the graph. One-way ANOVA (multiple comparison) was performed, and data are presented as mean values ± SD. **H** The CTDNEP1-NEP1R1-MAN1 complex dephosphorylates and inactivates R-SMADs at the INM (see text for details).

phosphatase CTDNEP1-NEP1R1. In this complex, MAN1 functions as a scaffold, bridging the interaction between the CTDNEP1-NEP1R1 phosphatase and phosphorylated R-SMADs, thereby promoting their dephosphorylation (Fig. 5G). The interaction with R-SMADs requires the MAN1 C-terminal nucleoplasmic domain, consistent with earlier studies[32,40,42]. MAN1 interaction with the phosphatase complex occurs via the regulatory subunit NEP1R1 and involves the transmembrane regions of both proteins. In particular, Alphafold predictions and mutagenesis analysis indicate that the first transmembrane segment of MAN1 interacts with both transmembrane domains of NEP1R1. Thus, besides a general role in stabilizing CTDNEP1[26], NEP1R1 is critical in SMAD regulation by promoting the interaction of the phosphatase complex with MAN1.

The regulation of other CTDNEP1-NEP1R1 substrates, such as Lipin[28] and the recently identified SUN2[30], does not appear to require MAN1. Instead, MAN1 functions as a specific and essential adapter for R-SMADs dephosphorylation. MAN1 exclusive localization to the INM ensures that R-SMAD dephosphorylation occurs only in the nucleus. These findings provide the mechanistic basis to explain previous results from mathematical modeling and time course experiments proposing that precise regulation of TGF-β signaling required R-SMADs to be dephosphorylated in the nucleus but not in the cytosol[7,53].

In active SMAD complexes, the phosphorylated SXS motif of one R-SMAD interacts with a basic pocket of a partner SMAD protein, either another R-SMAD or the co-SMAD SMAD4[54,55]. Thus, exposure of the SXS motif for dephosphorylation by the CTDNEP1-NEP1R1-MAN1 complex likely occurs in a coordinated fashion with SMAD complex disassembly. The monoubiquitination of SMADs, in particular, SMAD3 and SMAD4, was shown to stimulate the disassembly of SMAD complexes[56–58]. In the case of SMAD3, ubiquitination is mediated by the ubiquitin ligase SMURF2 and occurs at a lysine residue conserved in other R-SMADs, suggesting that this may be part of the general mechanism to disassemble SMAD complexes[58,59]. The mono-ubiquitination of SMAD4 involves a different ubiquitin ligase, TRIM33[56,57]. Curiously, the binding of TRIM33 to specific chromatin domains stimulates its ubiquitination activity towards SMAD4, providing an additional level of regulation of SMAD complex disassembly[60]. In the future, it will be interesting to test whether SMAD complex disassembly by monoubiquitination and CTDNEP1-NEP1R1-MAN1-dependent R-SMAD dephosphorylation are coordinated and if this potential coordination contributes to the temporal and spatial inactivation of SMAD signaling.

The binding of a TGF-β family cytokine to a cognate receptor at the surface of a target cell is known to initiate SMAD signaling[1,3]. We found that disruption of the CTDNEP1-NEP1R1-MAN1 complex not only delays SMAD inactivation upon ligand-induced stimulation but also triggers aberrant SMAD signaling in the absence of exogenous ligands. This observation suggests that, like other kinases, the intrinsic basal activity of TGF-β cytokine receptors is sufficient to phosphorylate and activate SMADs. Our data indicate that this activity is normally counteracted by the CTDNEP1-NEP1R1-MAN1 complex that constitutively dephosphorylate R-SMADs. In agreement with this model, we observe that the CTDNEP1-NEP1R1-MAN1 complex is present in cells and interacts with R-SMADs irrespective of TGF-β ligand stimulation. Moreover, the levels and composition of the CTDNEP1-NEP1R1-MAN1 complex appeared unchanged upon ligand-induced stimulation of TGF-β signaling, further supporting that it functions constitutively.

The mechanism of R-SMAD inactivation determined here also provides the molecular framework to understand the plethora of phenotypes caused by CTDNEP1 and MAN1 mutations in a variety of model organisms, such as flies[20,61–64], frogs[18,19,35,40] and mice[15–17,34,65–67]. Whole body deletion of CTDNEP1 is embryonic lethal in mice[14], while its conditional ablation leads to strong SMAD signaling deregulation in

various tissues, including kidney[15], bone[66], heart[17], and ovaries[67]. Moreover, MAN1 mutations result in a variety of disorders characterized by increased bone density, such as osteopoikilosis, Buschke-Ollendorff syndrome, and melorheostosis[36–39]. While the common denominator among these phenotypes is an increase in TGF-β/BMP signaling, the roles of CTDNEP1 and MAN1 remained enigmatic. This is now illuminated by our study. We also provide insight into the function of NEP1R1, for which there was less information aside from promoting CTDNEP1 stability[26,31].

Aggressive medulloblastomas frequently display CTDNEP1 mutations[21,22]. The loss of CTDNEP1 appears to potentiate the amplification of the MYC oncogene[23]. It was suggested that dephosphorylation of MYC Serine 62 by CTDNEP1 curbs its oncogenic activity. Interestingly, loss of CTDNEP1 also correlated with increased TGF-β signaling in these tumors. In the future, it will be interesting to explore if these two activities of CTDNEP1 are linked and how they contribute to medulloblastoma progression.

The role of CTDNEP1-NEP1R1 phosphatase in the regulation of Lipin has received great attention. Our study shows that this phosphatase has more pervasive functions in cell regulation. Other recent studies also suggest that CTDNEP1-NEP1R1 displays additional substrates[25,29,30]. In the future, it will be important to identify the complete substrate set of this phosphatase and understand its regulation.

## Methods

### Cells

HeLa cells were obtained from the ATCC. U2OS cells were obtained from the ECACC. The Lenti-X 293T cell line for the production of lentivirus was obtained from TakaraBio. Flp-In T-REx HEK293 cells were obtained from Invitrogen (Thermo Fischer Scientific). All cells were grown at 37 °C, 5% CO$_2$ in DMEM medium (Merck Life Science UK Limited #D6429) supplemented with L-Glutamine (2 mM; Gibco #25030024), Penicillin-Streptomycin (10 Units/mL; Gibco #15140122), and 10% FCS (Merck Life Science UK Limited #F9665).

For expressing recombinant CTDNEP1 proteins for the in vitro phosphatase assay, BL21-CodonPlus (DE3)-RIPL Competent cells (Agilent Technologies #230280) were cultured at 37 °C in LB media containing Kanamycin and chloramphenicol for plasmid transformation and outgrowth. For protein expression, cells were diluted in Terrific broth (Each 1L medium contains 12 g tryptone, 24 g yeast extract and 4 ml glycerol diluted in water), supplemented with K-Phos salt solution (170 mM KH2PO4 and 0.72 mM K2HPO4) and Kanamycin and grown at 37 °C until OD = 0.4–0.5, 0.4 mM Isopropyl β-d-1-thiogalactopyranoside (IPTG) were added to induce protein expression at 16 °C for 24 h.

### Plasmids

For overexpression and gene deletion of CTDNEP1, NEP1R1, MAN1, and PPM1A, cDNAs and sgRNAs were cloned in a dual promoter lentiviral vector as described in ref. 68. Constructs of CTDNEP1, CTDNEP1D67E, D69T mutant, NEP1R1, and PPM1A cDNAs were cloned into a lentiviral vector in which the protein expression was driven by the EF-1α promoter. Expression of MAN1, MAN1 mutants, and SMAD2 were doxycycline inducible and driven by cytomegalovirus (CMV) promoter. For recombinant CTDNEP1 protein expression used in the in vitro dephosphorylation assay, a vector containing H14-SUMO was used, with a lac operon to control protein expression by IPTG.

### Lentivirus production

For gene transductions using lentiviruses, the virus was produced using Lenti-X 293T cells in 24-well plates using TransIT LT-1 (Mirus Bio LLC #MIR 2305) and second-generation packaging vectors pMD2.G and psPAX2 according to standard lentiviral production protocols.

## Generation of CRISPR/Cas9-mediated knockout cells

Cell lines were transfected with a single bicistronic sgRNA-Cas9 plasmid using Mirus LT-1 according to the manufacturer's protocol. On the next day, cells were selected using Puromycin (2 μg/mL; Gibco #A1113803) for 72 h. To generate KO clones, cells were single-cell sorted using a BD AriaFusion flow cytometer. The knockout status of the clones was confirmed via immunoblotting and genomic sequencing. For genomic sequencing of KO clones, genomic DNAs (gDNA) were extracted from cells using QIAamp® DNA Blood Mini Kit (QIAGEN #51104) according to the manufacturer's protocol. PCR primers targeting ~250 bp upstream and downstream of the sgRNA cut site were used to amplify the purified gDNAs, resulting PCR products were analyzed by Sanger sequencing.

## Co-immunoprecipitation

Cells were lysed in 1% Decyl Maltose Neopentyl Glycol (DMNG) (Anatrace #NG322) lysis buffer (50 mM Tris-HCl pH 7.5, 150 mM NaCl) containing cOmplete EDTA-free protease inhibitor cocktail (Roche #5056489001) and phosphatase inhibitor cocktail (PhosSTOP, Roche #04906837001). Lysates were rotated at 4 °C for 120 min. Cell debris and nuclei were precipitated at $20,000 \times g$ at 4 °C for 20 min. Post-nuclear supernatants were incubated for 2 h with anti-HA magnetic beads (PierceTM, Thermo Fisher Scientific #88837), anti-FLAG magnetic beads (Sigma-Aldrich #M8823), or anti-V5 magnetic beads (MBL #M167-11). After three times 10 min washes in 0.1% DMNG washing buffer (50 mM Tris-HCl pH 7.5, 150 mM NaCl), proteins were eluted in 1× sample buffer for 15 min at 65 °C. The eluate was transferred to a new Eppendorf tube and subsequently reduced using dithiothreitol (DTT) (Merck Life Science UK Limited #D9779).

## Mass spectrometry-based analysis of immunoprecipitates

Triplicates of CTDNEP1- 3× FLAG and 3× V5-MAN1 and their corresponding empty vector control cell lines were subjected to cell lysis and co-immunoprecipitation as described in the previous session. When eluted, samples were resuspended in 25 μL 1× SDS sample buffer (5% SDS, 50 mM Triethylammonium bicarbonate (TAEB), pH 7.55). Disulfide bonds were reduced using 20 mM TCEP for 15 mins at 47 °C. After cooling down to room temperature, samples were alkylated using 20 mM CAA in the dark for 15 min, and 10% volume of 12% phosphoric acid was added to acidify the samples. S-trap binding buffer (90% methanol in 100 mM TAEB, pH 7.5) was added to acidified, denatured samples to a final volume of 190 μL, and the resulting solution was loaded onto S-Trap micro spin columns (ProtiFi), with a maximum of 150 μL of sample per load. Loaded spin columns were centrifuged at $4000 \times g$ for 1 min, and this step was repeated until the entire sample was loaded onto a spin column. S-Trap columns were washed 5× with S-trap binding buffer (90% methanol in 100 mM TAEB, pH 7.5), and the columns were transferred to 2 mL low-protein-binding Eppendorf tubes. For Trypsin/Lys-C digestion, 25 μL of digestion solution (50 mM TAEB, pH 8.0), containing 2 μg of Trypsin/Lys-C mix (Promega V5071) was added to each S-trap column, and the columns were incubated for 3 h at 47 °C on a ThermoMixer. Peptides were eluted with 30 μL of 50 mM TAEB, followed by 30 μL of 0.2% formic acid and 40 μL of 50% acetonitrile in 0.2% formic acid. Peptides were dried for 4 h at 37 °C in a vacuum centrifuge, and samples were stored at −80 °C until further analysis.

## Mass spectrometry analysis

Peptides were dissolved in 2% acetonitrile containing 0.1% trifluoroacetic acid, and each sample was independently analyzed on a Q Exactive HF Hybrid Quandrupole-Orbitrap mass spectrometer (Thermo Fisher Scientific), connected to an UltiMate 3000 RSLCnano System (Thermo Fisher Scientific). Peptides (1 μg) were injected on a PepMap 100 C18 LC trap column (300 μm ID × 5 mm, 5 μm, 100 Å) followed by separation on an EASY-Spray nanoLC C18 column (75 μm ID × 50 cm, 2 μm, 100 Å) at a flow rate of 250 nl min$^{-1}$. Solvent A was water containing 0.1% formic acid, and solvent B was 80% acetonitrile containing 0.1% formic acid. The gradient used for analysis of proteome samples was as follows: solvent B was maintained at 3% for 5 min, followed by an increase from 3 to 35% B in 60 min, 35%–90% B in 1 min, maintained at 90% B for 4 min, followed by a decrease to 3% in 0.5 min and equilibration at 2% for 10 min. The Q Exactive HF was operated in positive-ion data-dependent mode. The precursor ion scan (full scan) was performed in the Orbitrap in the range of 400–1500 m/z with a resolution of 120,000 at 200 m/z, an automatic gain control (AGC) target of $3 \times 10^6$, and an ion injection time of 50 ms. MS/MS spectra were acquired in high-energy collisional dissociation (HCD) mode with a normalized collision energy of 25 and resolution 15,000 using a Top 20 method, with a target AGC of $2 \times 10^5$ and a maximum injection time of 50 ms. The MS/MS triggering threshold was set at $5 \times 10^3$, and the dynamic exclusion of the previously acquired precursor was enabled for 45 s. For further LC and MS method information, please refer to the Source Data-proteomics file.

## Mass spectrometry data analysis

All spectra were analyzed using MaxQuant 1.6.10.43[69] and searched against a SwissProt homo sapiens fasta file (containing 42,371 database entries with isoforms, downloaded on 2021/02/24). Peak list generation was performed within MaxQuant, and searches were performed using default parameters and the built-in Andromeda search engine. The enzyme specificity was set to consider fully tryptic peptides, and two missed cleavages were allowed. Oxidation of methionine and N-terminal acetylation were allowed as variable modifications. Carbamidomethylation of cysteine was allowed as a fixed modification. Label-free quantification (LFQ) was enabled with a minimum ratio count of two. A protein and peptide false discovery rate (FDR) of less than 1% was employed in MaxQuant, and a minimum peptide length of 7 amino acids was accepted.

The search output files were further processed in R (version 4.4.0). First, all known contaminants, including bovine proteins from the cell medium, keratins from the researchers' skin and dust, as well as trypsin from enzymatic digestion, were removed from the protein list. Subsequently, proteins that were matched to the decoy database were removed from the list. Further, proteins identified by no unique peptide match and proteins that were only identified by a modified site were removed according to standard proteomics practices. A new protein groups column was generated from the Protein ID column, which contained solely the protein name for simplified labeling in the visualization. The intensity values from the filtered protein search output were log2 transformed and median normalized. In Perseus (version 2.0.3.1), proteins with 65% valid values across replicates per group were used, and missing values were imputed with "replace missing values from normal distribution" for statistical testing. A two-sided t-test was performed and the results plotted as volcano plot in R (significance cut-off: −1 and 1 log2 fold change, −log10 adj.p-value = 1.3).

## Immunoblotting

For immunoblotting, samples were incubated with 1× SDS sample buffer (1× sample buffer: 67 mM Tris-HCl (pH6.8), 2% SDS, 10% glycerol, 0.067% BromophenolBlue) with DTT at 65 °C for 10 min, separated by SDS-PAGE (Bio-Rad) and proteins were transferred to PVDF membranes (Bio-Rad). Membranes were blocked in 5% Milk or bovine serum albumin (BSA, Sigma, A9418) in phosphate-buffered saline (PBS, Thermo Fisher Scientific #D8537-500mL) containing 0.1% Tween20 (Sigma #P1379-500 mL) (PBS-Tween20) buffer and then probed with primary antibodies overnight at 4 °C on a shaker. After three washes with PBS buffer, secondary antibodies were performed at RT for 1 h either in 5% Milk or BSA in PBS-Tween20 buffer and subjected to another three washes with PBS. Membranes were developed by ECL

(Western Lightning ECL Pro, PerkinElmer) and visualized using an Amersham Imager 600 (GE Healthcare Life Sciences).

## Immunofluorescence assay

HeLa cells were seeded onto EprediaTM round coverslips (Thermo Scientific Menzel #17294914). Expression of wild-type and mutant MAN1 was induced with 1 μg/ml doxycycline for 24 h. For experiments requiring TGFβ treatment, 2 ng/mL TGFβ3 were treated for 1 h before fixation. Cells were then fixed with 4% methanol-free Paraformaldehyde (PFA) for 10 min at RT, followed by three PBS washes. Fixed cells were permeabilized in 0.2% Triton X-100 in PBS for 10 min, followed by 1 h blocking in blocking buffer (3% BSA, 0.2% Triton X-100 in PBS). Primary antibody cocktails were prepared in a blocking buffer. For primary antibody staining, coverslip was placed onto pre-spotted antibodies cocktail (40 μl/coverslip) on clean parafilm and incubated for 1 h underneath homemade aluminum foil-covered moisturized chamber. Coverslips were washed three times with blocking buffer, and secondary antibody staining was performed as was done for primary staining. Coverslips were then incubated in DAPI-containing PBS for 3 min, washed with PBS, and subsequently mounted onto glass slides in non-hardening mounting media and sealed with nail polish.

## Time course analysis of SMAD2 localization

To analyze SMAD2 localization, cells were incubated with 2 ng/mL TGF-β3 for 1 h, followed by PBS washing for three times before adding 10 μM SB431542 (cell signaling #14775) inhibitor, cells were fixed either before adding inhibitor or after 1 h, 3 h, 7 h and 19 h inhibitor treatment. Samples were processed for immunofluorescence as described above.

## Confocal fluorescence microscopy

Fixed cells on slides after immunofluorescence preparation were imaged using an inverted Zeiss 880 microscope fitted with an Airyscan detector using ZEN black software. The system was equipped with Plan-Apochromat ×63/1.4-NA oil lens, with an immersion oil (Immersol W 2010, Carl Zeiss; refractive index of 1.518). 488 nm argon and 405, 561, and 633 nm solid-state diode lasers were used to excite fluorophores. Z-sections of images were collected. The oil objective was covered with an immersion oil (ImmersolT 518F, Carl Zeiss) with a refractive index of 1.518. Microscopy images with CZI file format were analyzed using ImageJ (1.54fc, bundled with Java 1.8.0_172) software. Image quantification was performed by CellProfiler (4.2.1).

## pSMAD2 and pSMAD1 dephosphorylation kinetics assay

For testing the kinetics of pSAMD2 and pSAMD1 dephosphorylation, $8 \times 10^4$ cells were seeded for each time point on the previous day. The next day, cells were incubated with 2 ng/mL TGF-β3 (Cell signaling #8425 and #10858) or 20 ng/mL BMP-4 (Peprotech 120-05) for 1 h. After stimulation with different cytokines, cells were then subjected to PBS washing three times before adding 10 μM SB431542 (cell signaling #14775) or 0.1 μM LDN 193189 (Cambridge Bioscience HY-12071A −10mg) and incubated for different times. For SB431542 inhibitor, cells were collected after 30 min, 1 h, and 2 h treatment and lysed with 1× sample buffer. For LDN 193189 treatment, cells were lysed after 20 min, 40 min, and 60 min incubation. Cell lysates were separated with gel electrophoresis followed by immunoblotting analysis.

## Inhibition of TGF-β with 1D11 neutralizing antibody

Inhibition of TGF-β ligands was performed with the neutralizing antibody, 1D11 (BioX-Cell # #BE0083), and isotype-matched IgG1 monoclonal control antibody (BioX-Cell #BE0057) were used at 30, 150, and 300 μg/mL, respectively, to treat HeLa parental cells or CTDENP1 KO cells with indicated time. Cell lysates were separated with gel electrophoresis followed by immunoblotting analysis.

## RT-qPCR

Total RNA was extracted from HeLa Parental, PPM1A KO, MAN1 KO, CTDNEP1 KO, and NEP1R1 KO cell lines using Monarch® Total RNA Miniprep Kit (NEB T2010S). RT-qPCR was performed using the Luna Universal qPCR Master Mix (NEB M3003L) on a QuantStudio three Real-Time PCR System (Applied Biosystems) according to the manufacturer's instructions. Primers used for amplifying the specific regions of p21 and p15 are shown in Table S1. The individual-specific gene transcript levels in the KO cell lines are normalized to GAPDH transcriptional level and presented as fold change relative to HeLa parental control, determined using the comparative CT (ΔΔCT) method. All reactions were carried out in two to three biological replicates with each replicate analyzed in three technical replicates.

## Cell proliferation assay and flow cytometry

For EdU incorporation assay to assess cell proliferation, cells were seeded in 6-well plates for 24 h prior to 30 min incubation with 10 μM EdU (Abcam, #ab146186). Cells were then harvested, washed with PBS, and fixed with 70% Ethanol overnight. Fixed cells were pelleted by centrifuging at $1000 \times g$ for 5 min, washed with PBS, and permeabilized with PBSTri-BSA (PBS, 0.1% Triton X-100, 1% BSA) on ice for 15 min. Permeabilized cells were pelleted and washed twice with PBST-BSA (PBS, 0.1% Tween20, 1% BSA). EdU present in the cells were stained by Click-IT reaction (2 mM CuSO₄, 10 mM Sodium Ascorbate, 10 μM Alexa Fluor™ 555 Azide, Triethylammonium Salt (Thermo Fisher Scientific #A20012) in PBS) at room temperature for 1 h in the dark. After the Click-IT reaction, cells were pelleted and washed twice with PBST-BSA. DAPI staining was then performed in a DAPI-containing solution (PBS, 0.1% BSA, 1 mg/mL RNAse A (Thermo Fisher Scientific, # EN0531), 1 μg/mL DAPI (BD Bioscience, #564907)) for 1 h at room temperature in the dark. Next, cells were analyzed using a BD LSRFortessa X-20 flow cytometer. For each condition, 30,000 cells were measured, and FACS data was analyzed using FlowJo v10.

## Recombinant CTDNEP1 protein purification

Recombinant CTDNEP1 (46-244) wild-type and D67E, D69T mutant containing an N-terminal His14-Sumo tag and a C-terminal myc tag were expressed in BL21-CodonPlus(DE3)-RIPL Competent cells (Agilent Technologies #230280). Bacterial cells were grown in terrific broth (supplemented with K-Phos salt solution and Kanamycin) at 37 °C until OD600 = 0.4-0.5. Protein expression was then induced by 0.4 M Isopropyl β-d-1-thiogalactopyranoside (IPTG) at 16 °C for 24 h. Cells were harvested by centrifugation at 4000 rpm for 10 min and washed once with wash buffer (50 mM Tris-HCl (pH 8.0), 500 mM NaCl, 30 mM imidazole, 1 mM PMSF, 1.8 mM PepA). Cells were lysed by 1 mg/ml lysozyme + .05 mg/ml DNaseI in WB, incubating at room temperature for 30 min. After incubation, cell lysates were sonicated on ice by a probe sonicator (Soniprep 150), with amplitude set at 10–15 microns and on/off pulse of 30 s duration for 5 times. The lysate was cleared by centrifugation at 4000 rpm for 10 min. Cleared lysates were subjected to ultracentrifugation in Ti45 tubes at 40,000 rpm for 45 min at 4 °C to separate a crude membrane fraction. Supernatants were collected and incubated with Ni-NTA Agarose beads (Thermo Scientific, HisPurTM #88222) overnight at 4 °C. After incubation, the material was transferred to a 20 ml gravity column, and beads were washed by gravity flow with 20 column volumes of WB. Proteins that were bound on the Ni-NTA beads were eluted by elution buffer (50 mM Tris-HCl (pH 8.0), 500 mM NaCl, 300 mM imidazole). Eluted protein was pooled and loaded onto a Superdex 200 Increase 10/300 GL column (GE #28-9909-44) equilibrated with buffer (20 mM HEPES (pH 7.4), 200 mM NaCl). Peak fractions were collected and snap frozen until use. To obtain His-SUMO protein as the control of the in vitro phosphatase, an aliquot of purified His-SUMO-CTDNEP1 WT was immobilized on Ni-NTA beads in 10 mL of WB for 1 h at 4 °C. 1 μM of Ulp1 was added to the beads and incubated further for 2 h at 4 °C.

The material was transferred to a 20 mL gravity column, and the beads were washed by gravity flow with 10 column volumes of WB. His-SUMO protein bound on the Ni-NTA beads was eluted by elution buffer and snap frozen.

### In vitro phosphatase assay

Lentiviral plasmid encoding SMAD2 with N-terminal 3× HA tag was transduced into a HeLa MAN1 and CTDNEP1 double KO clone. Cells were treated with 1 µg/ml doxycycline for 24 h to induce HA-SMAD2 expression, followed by 20 ng/mL TGFβ3 treatment for 1 h to obtain a pool of phospho-SMAD2. Cells were lysed with 1% DMNG followed by anti-HA immunoprecipitation as described in the previous method section. In vitro dephosphorylation assay was applied to HA-pSMAD2 beads with recombinantly expressed and purified 100 ng and 1 µg of wild-type or catalytically inactive soluble CTDNEP1 at 37 °C for 10 min. For quenching the assay, the beads were incubated with 1× SDS sample buffer at 65 °C for 20 min.

### Western blot quantification

Western blot data was quantified using Image Studio software (Li-Cor Ver5.2), and graphs were plotted using Prism (GraphPad). Representative images of at least three independent experiments are shown. Student $t$-test was used for statistical analysis, and error bars represent the standard deviation.

### Quantification of SMAD2 localization

Nucleo:Perinuclear ratio of SMAD2 intensity was quantified by Cell-Profiler (4.2.1). Raw microscopic images of SMAD2 and DAPI were imported to ImageJ (1.54f, bundled with Java 1.8.0_172) software to acquire maximum projection of the z-stack images. Maximum projection image files of SMAD2 and DAPI were imported to CellProfiler. DAPI staining was used as the indicator of the nuclear area, and SMAD2 staining was used to indicate the whole cell area. The cytoplasmic area was determined by subtracting the nuclear area from the whole cell area. Within the cytoplasmic area, the perinuclear area was defined by a 10-pixel ring surrounding the nucleus. Integrated intensity of SMAD2 in the nuclear area and perinuclear area was measured and the ratio was calculated in logarithms for statistical analysis. One-way ANOVA test and graph plotting were performed using GraphPad Prism 10. For each sample, more than 100 cells were analyzed, which were obtained from at least three independent experiments and at least three different fields per sample from each experiment. Error bars represent the standard error of the mean of three replicates.

### Quantification of EdU staining

Flow cytometry data was analyzed by FlowJo v10. For each sample, the EdU-positive cells were gated from a total of 30,000 cells, and the percentage of EdU+ cells from three independent replicates were compared between control cells and the KO cells in GraphPad Prism 10 using one-way ANOVA. Error bars represent the standard deviation.

### Statistics and reproducibility

Western blot data was quantified using Image Studio software (Li-Cor), and graphs were plotted using Prism (GraphPad). Representative images of at least three independent experiments are shown. Error bars represent the standard deviation, and the measure of center represents the mean.

### Reporting summary

Further information on research design is available in the Nature Portfolio Reporting Summary linked to this article.

## Data availability

The mass spectrometry proteomics data have been deposited to the ProteomeXchange Consortium via the PRIDE[70] partner repository with the dataset identifier PXD051056. The published article includes all processed data generated or analyzed during this study as Supplemental Information. Any additional information required to reanalyze the data reported in this paper is available from the lead contact upon request. Source data are provided with this paper.

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

## Acknowledgements
We thank C. Hill for discussions and U. Gruneberg, C. Hill, and R. Klemm for critical reading of the manuscript. PC was supported by an investigator award from the Wellcome Trust (223153/Z/21/Z) and an ERC consolidator grant (GA 817708). This research was co-funded by grant awards to MT (Wellcome Trust Multi-User Equipment grant (212947/Z/18/Z) and Investigator Award (215542/Z/19/Z)).

## Author contributions
Z.J. and W.S.S. performed all the experiments. Z.J., W.S.S., and P.C. analyzed the data. M.E.D., L.M., and M.T. generated and analyzed the mass spectrometry data. P.C. conceived and supervised the project and wrote the manuscript with input from all the authors.

## Competing interests
The authors declare no competing interests.
