## [Transparent Peer Review file · Nature Communications]

Suppression of TGF- β /SMAD signaling by an inner nuclear membrane phosphatase complex

Corresponding Author: Dr Pedro Carvalho

Version 0:

Reviewer comments:

Reviewer #1

(Remarks to the Author)

In this paper, the CTDNEP1-NEP1R1 phosphatase complex is shown to dephosphorylate R-SMADs. MAN1 is shown to be part of that complex and forms a bridge between CTDNEP1-NEP1R1 and R-SMADs (interacting independently by these components). The results support the conclusions, albeit some specific experiments need further validation and clarification. The approaches are confined to in vitro cell culture, biochemical/proteomic experiments, and genetic and pharmacological approaches.

Specific comments:

1. This paper builds on previous reports that have shown that (1) NEP1R1/Dullard inhibits R-SMAD C-terminal phosphorylation (DOI: 10.1038/srep32269, DOI: 10.7554/eLife.50325; DOI: 10.1002/jbmr.2343), (2) MAN1/LEMD3 interacts with TGF-beta and BMP R-SMADs (DOI: 10.1093/nar/gky925), inhibits TGF- β /SMAD2/3 signaling (DOI: 10.1074/jbc.RA118.003658; DOI: 10.1074/jbc.M411234200; DOI: 10.1242/dev.00401; DOI: 10.1074/jbc.M210505200) and (3) CTDNEP1 and NEP1R1 interact (DOI: 10.1073/pnas.2321167121). The authors need to clarify/emphasize what is new in the present manuscript: the interaction of MAN1 with CTDNEP1 and the formation of a scaffold for R-SMAD recruitment.
2. At the end of the introduction, it is stated that CTDNEP1-NEP1R1 is identified as the long-sought phosphatase that dephosphorylates TGF-beta and BMP R-SMADs. Considering previous reports of NEP1R1/Dullard inhibiting R-SMAD C-terminal phosphorylation (DOI: 10.1038/srep32269, DOI: 10.7554/eLife.50325; DOI: 10.1002/jbmr.2343), I find this an overstatement. Also, other phosphatases have been shown to target C-terminal phosphorylate R-SMADs (DOI: 10.1074/jbc.M109.075036; DOI: 10.1101/gad.1384706; DOI: 10.1016/j.cell.2006.03.044; DOI: 10.1074/jbc.M607246200). The authors find that PPM1A, previously reported to act as R-SMAD Terminal phosphatase, is inactive using their experimental conditions. Can it be excluded that in other cells/conditions, it might function as a phospho-R-SMAD phosphatase?
3. CTDNEP1/Dullard has been shown to dephosphorylate (and induce degradation of) BMP receptors (DOI: 10.1016/j.devcel.2006.10.001) (BMPR act upstream of phosphorylated R-SMAD1/5). Can CTDNEP1 act on phosphorylated TGF-beta receptors? And thereby contribute to the depletion of CTDNEP1, NEP1R1, or MAN1- induced SMAD2 nuclear accumulation without exogenous TGF-beta stimulation?
4. The paper is focused on TGF-beta signaling, but CTDNEP1/Dullard has also been shown to target BMP receptors and BMP RSMAD signaling. BMP receptor/SMAD signaling interacts at multiple levels with TGF-beta receptor/SMAD signaling. Could some of the effects on TGF-beta signaling observed upon mis-expression of CTDNEP1 lead to indirect effects of TGF-beta R-SMAD signaling responses?
5. Figure 5. Neutralizing antibodies against TGF-beta are less effective than treatment with SB431542. Could this point to a role of activin (or nodal), whose receptor signaling is also inhibited by SB431542 but not by TGF-beta neutralizing antibody? SB431542 should not be indicated as a selective TGF-beta receptor kinase inhibitor.

Reviewer #2

(Remarks to the Author)

In this paper, the authors performed biochemical assays, immunoprecipitation, mass spectrometry, and microscopy imaging analyses to demonstrate that TGF- β /SMAD signaling is inhibited through dephosphorylation mediated by CTDNEP1-NEP1R1 phosphatase activity. My review of this manuscript is based on my expertise in mass spectrometry, co-immunoprecipitation, Western blot analysis, and other biochemical techniques, as well as the data presented in the manuscript. However, I am not an expert in cytokine biology or the role of TGF- β in physiological and stress-related contexts. The mass spectrometry results (Table S1) show that hundreds of other proteins were detected in the co-IP/MS samples, with over 70 proteins showing significant differences between CTDNEP1 WT_EV IP-MS. The numbers are even higher for MAN1_EV IP-MS. The authors claimed that "Importantly, interactions between the CTDNEP1-NEP1R1 phosphatase complex and MAN1 appeared specific since SUN1 and other abundant INM proteins were not present in the precipitate, as assayed by immunoblot (Fig. 1B-C, S1D-E) and mass spectrometry."

If the interaction between CTDNEP1-NEP1R1 and MAN1 is specific, how do the authors interpret the presence of dozens of other proteins that show significant enrichment? Below I have several specific comments, and I think addressing these comments would help to improve the quality of this manuscript.

First, it would be helpful if the authors provided the original peptide and protein output tables from the MaxQuant search (apologies if I missed them). It was unclear how proteins specifically pulled down in CTDNEP1 WT-3FLAG and V5-MAN1 WT were filtered from those identified in the empty vector control. While the provided tables include log-transformed intensity and iBAQ values, I could not find information on MS/MS spectral counts for all identified and quantified peptides across the samples. The MS/MS counts provided are cumulative for all samples, but it would be beneficial to see how these counts are distributed across the six individual samples. This information is critical for assessing the confidence in protein identifications since MS/MS counts provide sequence-level validation.

Could the authors justify why iBAQ was used for comparison instead of LFQ?

To my understanding, iBAQ (intensity-based absolute quantification) normalizes peptide and protein intensities (abundances) within a sample based on protein size (molecular weight), whereas LFQ (label-free quantification) normalizes data across all samples. Given that statistical comparisons were performed between co-IP samples, it would be more appropriate to use LFQ values rather than iBAQ values.

Additionally, original peptide output tables after MaxQuant search should be provided.

Detailed information for bioinformatic and statistical analysis should be provided. Did they use Perseus? If yes, please provide details about Perseus analysis.

The authors mentioned that two unique peptides were required to identify a protein; however, Table S1 does not clarify this filtering criterion. It would also be beneficial to explain how unique and shared peptides and proteins were managed and analyzed. For example, many significant proteins listed include multiple protein accession numbers—clarification on how these protein groups were handled is necessary.

Finally, it is essential to provide a detailed description of the mass spectrometry parameters used, as this would help assess the reliability and reproducibility of the results.

The authors performed a series of co-IP experiments to validate their findings; however, I suggest complementing these results with additional experiments. While untargeted mass spectrometry analysis offers valuable insights, incorporating targeted proteomics analyses could further strengthen the conclusions.

Overall, while the biochemical and mass spectrometry analyses were well-designed and executed, the findings contribute only a modest advance to the existing knowledge in the field.

Reviewer #3

(Remarks to the Author)

This beautifully written and convincing manuscript solves a long-standing mystery: the mechanism by which a nuclear membrane protein, MAN1, dampens/regulates cytokine signaling by activated transcription factors (SMADs) that have already entered the nucleus. This seemingly 'rogue' function of a nuclear membrane protein defied the canonical understanding of signaling.

This manuscript provides convincing molecular evidence that MAN1 at the inner nuclear membrane serves as a 'bridge' that connects SMADs (known to bind the C-terminus of MAN1) with a membrane-anchored phosphatase complex, by associating with bilayer-spanning domains of the regulatory subunit (NEP1R1) and its catalytic subunit (CTDNEP1).

Although not mentioned in this manuscript, these findings have groundbreaking implications for other conserved nuclear envelope proteins, especially emerin and lamins:

(a) Emerin (the "E" in LEM-domain) is required to 'expel' activated messengers in the Wnt and MAPK pathways, via unknown mechanisms. Emerin can bind the N-terminal domain of MAN1, and also co-IP'd with CTDNEP1 in this work (line 64 in Table S1).

(b) A-type lamins are major influencers of tissue-specific signaling and gene expression. A- and especially B-type lamins can bind the N-terminal domain of MAN1 and also co-IP'd with CTDNEP1 in this work (LMNA on line 429; LMNB1 on line 480 of Table S1; take with 'grain-of-salt' due to insolubility).

Other items to improve clarity or accuracy:

Line 126: MAN1 is understudied, not "poorly characterized".

Line 131: typo "CTDNEP1D67E,D69T"

Figure 1 legend: Define “SE” and “LE” here, and Results text.

Line 136-137: Unclear. Change to “Nep1R1-HA also co-precipitated..”

Lines 203-205: Rephrase to describe PPM1A’s function more accurately. E.g., ‘indicating that PPM1A function is limited to... [what, specifically]?’

Line 209: Dephosphorylation activity was detected in Fig 2E, but was not convincingly “efficient”, since some SMAD2 remains phosphorylated despite microgram amounts of an enzyme.

Lines 208-209: Please specify that the recombinant protein purified from bacteria was a soluble fragment comprising residues 46-244 of CTDNEP1.

Lines 242-243: This statement (“binding of NEP1R1 to MAN1 was only slightly reduced in CTDNEP1-KO cells”) is unconvincing: reduction was not quantified, and does not appear to be ‘slight’.

Lines 251-253: LEMD2 is even more understudied than MAN1; wise to avoid the implication that it only has one role.

Fig 4F legend: Define ‘EV’.

Lines 291-292 (Fig 5D): Name the ligand and inhibitor early in the results section to facilitate direct interpretation of Figure 5D. Legend line 731 (‘indicated treatments’) does not name the target or the consequences of antibody 1D11, nor is “SB431542” explained.

Lines 335-336 and the model in Figure 5H: the figure suggests that the two transmembrane domains of NEP1R1 both associate with the second transmembrane domain of MAN1. If this is accurate, please state explicitly on line 336. Would also be helpful to briefly summarize the molecular interaction between CTDNEP1 and NEP1R1 in the text.

Lines 335-336. Figure 5H depicts CTDNEP1 as a peripheral (non-integral) membrane protein, not an integral membrane protein as strongly implied on lines 77-80. To avoid confusion, please describe their membrane-associations accurately in the introduction.

Version 1:

Reviewer comments:

Reviewer #1

(Remarks to the Author)

The authors have improved the clarity of the manuscript by textual changes and addition of extra (available) information.

Reviewer #2

(Remarks to the Author)

Thank you for your satisfactory response and revision of the manuscript based on my previous comments.

Reviewer #3

(Remarks to the Author)

This revised and improved manuscript, and revised Fig 5H, fully address my previous concerns. I'm happy the authors plan to follow up on emerin, though I still feel its major disease-relevant implications deserve mention here for readers outside the nuclear envelope field.

This manuscript is a major advance in cell biology because it solves a long-standing mystery: the mechanism by which (at least) one activated signaling pathway is controlled by lamin-associated regulatory complexes at the inner nuclear membrane.

Just two small revisions related to the NE as an ER-adjacent but nonetheless unique compartment:

Line 92: change “small ER membrane” to “small ER/NE membrane”.

Line 110: change “the ER, including the nuclear envelope” to “the ER and nuclear envelope”.

Response to the reviewers:

Reviewer #1 (Remarks to the Author):

In this paper, the CTDNEP1-NEP1R1 phosphatase complex is shown to dephosphorylate R-SMADs. MAN1 is shown to be part of that complex and forms a bridge between CTDNEP1-NEP1R1 and R-SMADs (interacting independently by these components). The results support the conclusions, albeit some specific experiments need further validation and clarification. The approaches are confined to in vitro cell culture, biochemical/proteomic experiments, and genetic and pharmacological approaches.

Specific comments:

1. This paper builds on previous reports that have shown that (1) NEP1R1/Dullard inhibits R-SMAD C-terminal phosphorylation (DOI: 10.1038/srep32269, DOI: 10.7554/eLife.50325; DOI: 10.1002/jbmr.2343, (2) MAN1/LEMD3 interacts with TGF-beta and BMP R-SMADs (DOI: 10.1093/nar/gky925), inhibits TGF-b/SMAD2/3 signaling (DOI: 10.1074/jbc.RA118.003658; DOI: 10.1074/jbc.M411234200; DOI: 10.1242/dev.00401; DOI: 10.1074/jbc.M210505200) and (3) CTDNEP1 and NEP1R1 interact (DOI: 10.1073/pnas.2321167121). The authors need to clarify/emphasize what is new in the present manuscript: the interaction of MAN1 with CTDNEP1 and the formation of a scaffold for R-SMAD recruitment.

We thank the reviewer for raising this point. We have now substantially changed the text to highlight the new discoveries described in our manuscript.

As previously showed in flies we now demonstrate that CTDNEP1 dephosphorylates -SXS motif of R-SMADs in human cells. Moreover, we showed for the first time that this dephosphorylation reaction requires NEP1R1; that MAN1 is a substrate specific adaptor for the CTDNEP1/NEP1R1 complex at the inner nuclear membrane; that CTDNEP1/NEP1R1/MAN1-dependent dephosphorylation of R-SMADs is important not only to suppress signaling upon ligand stimulation of the pathway but also to prevent basal activity.

2. At the end of the introduction, it is stated that CTDNEP1-NEP1R1 is identified as the long-sought phosphatase that dephosphorylates TGF-beta and BMP R-SMADs. Considering previous reports of NEP1R1/Dullard inhibiting R-SMAD C-terminal phosphorylation (DOI: 10.1038/srep32269, DOI: 10.7554/eLife.50325; DOI: 10.1002/jbmr.2343), I find this an overstatement. Also, other phosphatases have been shown to target C-terminal phosphorylate R-SMADs (DOI: 10.1074/jbc.M109.075036; DOI: 10.1101/gad.1384706; DOI: 10.1016/j.cell.2006.03.044; DOI: 10.1074/jbc.M607246200). The authors find

that PPM1A, previously reported to act as R-SMAD Terminal phosphatase, is inactive using their experimental conditions. Can it be excluded that in other cells/conditions, it might function as a phospho-R-SMAD phosphatase Can it be excluded that in other cells/conditions, it might function as a phospho-R-SMAD phosphatase?

As mentioned in the previous point, we have changed the text and directly acknowledge previous work in flies.

The fact that PPM1A KO mice are viable and do not exhibit phenotypes consistent with deregulated TGF- β /SMAD signaling strongly suggests that PPM1A does not play a significant role in R-SMAD dephosphorylation. However, we cannot formally rule out the possibility that PPM1A may dephosphorylate R-SMADs under specific conditions or in certain cell types. We believe we describe this in a balanced manner in our manuscript.

3. CTDNEP1/Dullard has been shown to dephosphorylate (and induce degradation of) BMP receptors (DOI: 10.1016/j.devcel.2006.10.001) (BMPR act upstream of phosphorylated R-SMAD1/5). Can CTDNEP1 act on phosphorylated TGF-beta receptors? And thereby contribute to the depletion of CTDNEP1, NEP1R1, or MAN1- induced SMAD2 nuclear accumulation without exogenous TGF-beta stimulation?

In this manuscript we focused on how the CTDNEP1/NEP1R1 phosphatase and the INM protein MAN1 cooperate to dephosphorylate R-SMADs in the nucleus. While CTDNEP1 may contribute to BMP/TGF- β signaling through additional mechanisms, including the one raised by the reviewer, exploring these hypotheses falls outside the scope of this study.

As a side note, we would like to point out that our thorough mass spectrometry analysis did not identify BMP or TGF- β receptors among the interactors of CTDNEP1. In contrast, endogenous NEP1R1 and MAN1 were the strongest associated proteins.

4. The paper is focused on TGF-beta signaling, but CTDNEP1/Dullard has also been shown to target BMP receptors and BMP RSMAD signaling. BMP receptor/SMAD signaling interacts at multiple levels with TGF-beta receptor/SMAD signaling. Could some of the effects on TGF-beta signaling observed upon mis-expression of CTDNEP1 lead to indirect effects of TGF-beta R-SMAD signaling responses?

We show that the function of CTDNEP1/NEP1R1/MAN1 is critical to suppress all R-SMAD-dependent signaling downstream of both TGF- β and BMP pathways. While data on TGF- β elicited signaling are presented in multiple figures, Figure 2C specifically shows that CTDNEP1/NEP1R1/MAN1 function is critical for R-SMAD dephosphorylation upon BMP stimulation.

Moreover, we observed that CTDNEP1/NEP1R1/MAN1 associates with all R-SMADs, irrespective if they mediate TGF- β (SMAD2/3) or BMP (SMAD1/5/8) signaling- see Figures 1, 3, 4 and S1. This observation is consistent with earlier structural data (DOI: 10.1093/nar/gky925) showing that SMAD1 (phosphorylated in response to BMP) and SMAD2 (phosphorylated in response to TGF- β) share a similar interface and binding mode for interacting with MAN1 and, by extrapolation, for dephosphorylation by CTDNEP1/NEP1R1.

In summary, we propose that the mechanism of R-SMAD inactivation described here is universal across TGF- β and BMP signaling pathways.

5. Figure 5. Neutralizing antibodies against TGF-beta are less effective than treatment with SB431542. Could this point to a role of activin (or nodal), whose receptor signaling is also inhibited by SB431542 but not by TGF-beta neutralizing antibody? SB431542 should not be indicated as a selective TGF-beta receptor kinase inhibitor.

We thank the reviewer for pointing this out. Using RT-qPCR, we did not detect expression of *Nodal* or *Activin* in the various HeLa cell lines used in the study (parental, MAN1 KO, CTDNEP1 KO, NEP1R1 KO). This is in line with extensive literature indicating that Nodal is not normally expressed in adult tissues (see for example this review article doi: 10.1016/j.ceb.2017.10.005.). As discussed in the manuscript, we favor the idea that the different effectiveness between the 1D11 neutralizing antibody and the receptor kinase inhibitor SB431542 reflects a basal activity of the receptor, which is normally counteracted by the CTDNEP1/NEP1R1/MAN1 function and that is defective in the various KO cell lines.

Reviewer #2 (Remarks to the Author):

In this paper, the authors performed biochemical assays, immunoprecipitation, mass spectrometry, and microscopy imaging analyses to demonstrate that TGF- β /SMAD signaling is inhibited through dephosphorylation mediated by CTDNEP1-NEP1R1 phosphatase activity. My review of this manuscript is based on my expertise in mass spectrometry, co-immunoprecipitation, Western blot analysis, and other biochemical techniques, as well as the data presented in the

manuscript. However, I am not an expert in cytokine biology or the role of TGF- β in physiological and stress-related contexts.

The mass spectrometry results (Table S1) show that hundreds of other proteins were detected in the co-IP/MS samples, with over 70 proteins showing significant differences between CTDNEP1 WT_EV IP-MS. The numbers are even higher for MAN1_EV IP-MS. The authors claimed that “Importantly, interactions between the CTDNEP1-NEP1R1 phosphatase complex and MAN1 appeared specific since SUN1 and other abundant INM proteins were not present in the precipitate, as assayed by immunoblot (Fig. 1B-C, S1D-E) and mass spectrometry.”

If the interaction between CTDNEP1-NEP1R1 and MAN1 is specific, how do the authors interpret the presence of dozens of other proteins that show significant enrichment? Below I have several specific comments, and I think addressing these comments would help to improve the quality of this manuscript.

There is vast literature demonstrating that the inner nuclear membrane (INM) of mammalian cells contains many dozens of membrane proteins. Among these, MAN1 is the only INM protein that is significantly enriched in the CTDNEP1 IPs, as detected by mass spectrometry and analysed using Maxquant. We find this a remarkably specific association. AlphaFold structural predictions allowed us to identify the mode of association between CTDNEP1 and MAN1 that involved NEP1R1, an ER membrane protein that is also specifically enriched in CTDNEP1 precipitates.

Our mass spectrometry analysis indicates that CTDNEP1 has additional partners. However, in this manuscript we focus specifically on its interaction with MAN1 and the critical role of this interaction for SMAD dephosphorylation. We believe this is clear in our manuscript.

It was unclear how proteins specifically pulled down in CTDNEP1 WT-3FLAG and V5-MAN1 WT were filtered from those identified in the empty vector control.

To determine the proteins of interest from the pulldown, a quantitative proteomics experiment between the sample and an empty vector control was performed. The resulting protein fold changes and p-values were displayed in a volcano plot to identify significantly changed proteins between the conditions. The highlighted proteins were either not present in the vector control (ratios obtained by imputation) or at much lower levels as the very high fold-changes show. Next to the identified CNEP1, NEP1R1, MAN1 and SMADs, a large portion of significant hits were ribosomal proteins, commonly identified by MS as contaminants of IPs.

First, it would be helpful if the authors provided the original peptide and protein output tables from the MaxQuant search (apologies if I missed them). While the provided tables include log-transformed intensity and iBAQ values, I could not find information on MS/MS spectral counts for all identified and quantified peptides across the samples. The MS/MS counts provided are cumulative for all samples, but it would be beneficial to see how these counts are distributed across the six individual samples. This information is critical for assessing the confidence in protein identifications since MS/MS counts provide sequence-level validation.

The original MaxQuant search protein output file can be found in the PRIDE Archive and this file contains the cumulative MS/MS counts. We thank the reviewer for the suggestion and added the msms search output file to the Source Data file for proteomics to enable more detailed insight into the MS/MS counts per peptide per sample.

Could the authors justify why iBAQ was used for comparison instead of LFQ? To my understanding, iBAQ (intensity-based absolute quantification) normalizes peptide and protein intensities (abundances) within a sample based on protein size (molecular weight), whereas LFQ (label-free quantification) normalizes data across all samples. Given that statistical comparisons were performed between co-IP samples, it would be more appropriate to use LFQ values rather than iBAQ values.

We thank the reviewer for this comment. We conducted the analysis with the intensity values and updated the figures accordingly. We would like to point out that even though significance levels and fold changes altered slightly, the top significant proteins remained the same. For further validation, we re-analysed our data with the FragPipe search engine which revealed upon analysis of the LFQ values a similar list of highly significant proteins.

Additionally, original peptide output tables after MaxQuant search should be provided.

We thank the reviewer for pointing out that the original peptide search file was missing, and we have rectified this by uploading this file as part of our Source Data file for proteomics.

Detailed information for bioinformatic and statistical analysis should be provided. Did they use Perseus? If yes, please provide details about Perseus analysis.

We would like to thank the reviewer for this comment. We now provided more detail in the data analysis method section as requested.

The authors mentioned that two unique peptides were required to identify a protein; however, Table S1 does not clarify this filtering criterion. It would also be beneficial to explain how unique and shared peptides and proteins were managed and analyzed. For example, many significant proteins listed include multiple protein accession numbers—clarification on how these protein groups were handled is necessary.

We thank the reviewer for spotting that we should have written one unique peptide was required to identify a protein. We corrected this in the methods section. By using the original search output file and the updated supplementary table, readers should now be able to more easily follow the data processing workflow.

We would like to clarify that the MaxQuant search output intensities that were used for the analysis are by default the sums of unique and razor peptide intensities per protein group and that as described in the method section those default settings were used.

Lastly, we would like to address the significant proteins that have multiple accessions in the protein column. A closer look into these accessions revealed that mostly proteins in their indistinguishable isoforms were annotated in this way. To simplify the visualization, protein names were extracted from the Protein ID column to distinguish between confident protein identification (irrespective of the isoform) and ambiguous protein identification. Different technologies and workflows exist to distinguish between isoforms, however this was beyond the scope of this manuscript.

Finally, it is essential to provide a detailed description of the mass spectrometry parameters used, as this would help assess the reliability and reproducibility of the results.

The mass spectrometry analysis section was adjusted and an additional file with further method information (LS and MS) is now available in the Source Data file for proteomics.

The authors performed a series of co-IP experiments to validate their findings; however, I suggest complementing these results with additional experiments. While untargeted mass spectrometry analysis offers valuable insights, incorporating targeted proteomics analyses could further strengthen the conclusions.

We thank the reviewer for the suggestion. The primary focus of our manuscript is to explore the role of CTDNEP1 in SMAD signaling. We believe our data unequivocally demonstrate that CTDNEP1 associates with MAN1 and highlight the importance of CTDNEP1/NEP1R1/MAN1 in SMAD dephosphorylation. We believe that incorporating additional targeted proteomics would not provide substantial new insights to our study.

Overall, while the biochemical and mass spectrometry analyses were well-designed and executed, the findings contribute only a modest advance to the existing knowledge in the field.

We thank the reviewer for acknowledging the quality of our data. However, we respectfully disagree the assessment. Our study shows for the first time that CTDNEP1 dephosphorylates R-SMADs, a reaction that also requires NEP1R1. Moreover, we discovered that MAN1 is a substrate specific adaptor for the CTDNEP1/NEP1R1 complex at the inner nuclear membrane. Importantly, we show that CTDNEP1/NEP1R1/MAN1-dependent dephosphorylation of R-SMADs is important not only for suppressing signaling upon ligand stimulation of the pathway but also for preventing basal pathway activity.

Considering the importance of SMAD dependent gene regulation during both development and adulthood as well as its close links to human disease, we believe our findings provide a step change in understanding SMAD regulation. In fact, this is also the opinion of both reviewers 1 and 3.

Reviewer #3 (Remarks to the Author):

This beautifully written and convincing manuscript solves a long-standing mystery: the mechanism by which a nuclear membrane protein, MAN1, dampens/regulates cytokine signaling by activated transcription factors (SMADs) that have already entered the nucleus. This seemingly 'rogue' function of a nuclear membrane protein defied the canonical understanding of signaling.

This manuscript provides convincing molecular evidence that MAN1 at the inner

nuclear membrane serves as a ‘bridge’ that connects SMADs (known to bind the C-terminus of MAN1) with a membrane-anchored phosphatase complex, by associating with bilayer-spanning domains of the regulatory subunit (NEP1R1) and its catalytic subunit (CTDNEP1).

Although not mentioned in this manuscript, these findings have groundbreaking implications for other conserved nuclear envelope proteins, especially emerin and lamins:

(a) Emerin (the “E” in LEM-domain) is required to ‘expel’ activated messengers in the Wnt and MAPK pathways, via unknown mechanisms. Emerin can bind the N-terminal domain of MAN1, and also co-IPd with CTDNEP1 in this work (line 64 in Table S1).

(b) A-type lamins are major influencers of tissue-specific signaling and gene expression. A- and especially B-type lamins can bind the N-terminal domain of MAN1 and also co-IP’d with CTDNEP1 in this work (LMNA on line 429; LMNB1 on line 480 of Table S1; take with ‘grain-of-salt’ due to insolubility).

We thank the reviewer for highlighting the importance of our findings. We agree that there is some additional interesting information in our mass spectrometry experiments. We plan to follow up on some of the results in the near future.

Other items to improve clarity or accuracy:

Line 126: MAN1 is understudied, not “poorly characterized”.

Thanks for the suggestion. This has been fixed.

Line 131: typo “CTDNEP1D67E,D69T”

This has been fixed.

Figure 1 legend: Define “SE” and “LE” here, and Results text.

This has been fixed.

Line 136-137: Unclear. Change to “Nep1R1-HA also co-precipitated..”

This has been fixed.

Lines 203-205: Rephrase to describe PPM1A's function more accurately. E.g., 'indicating that PPM1A function is limited to...[what, specifically]?'

We changed the sentence to improve clarity.

Line 209: Dephosphorylation activity was detected in Fig 2E, but was not convincingly "efficient", since some SMAD2 remains phosphorylated despite microgram amounts of an enzyme.

This has been fixed.

Lines 208-209: Please specify that the recombinant protein purified from bacteria was a soluble fragment comprising residues 46-244 of CTDNEP1.

This has been fixed.

Lines 242-243: This statement ("binding of NEP1R1 to MAN1 was only slightly reduced in CTDNEP1-KO cells") is unconvincing: reduction was not quantified, and does not appear to be 'slight'.

The text has been changed to more accurately describe the data.

Lines 251-253: LEMD2 is even more understudied than MAN1; wise to avoid the implication that it only has one role.

This has been fixed.

Fig 4F legend: Define 'EV'.

This has been fixed.

Lines 291-292 (Fig 5D): Name the ligand and inhibitor early in the results section to facilitate direct interpretation of Figure 5D. Legend line 731 ('indicated treatments') does not name the target or the consequences of antibody 1D11, nor is "SB431542" explained.

We changed the text and figure legend to make this clearer.

Lines 335-336 and the model in Figure 5H: the figure suggests that the two transmembrane domains of NEP1R1 both associate with the second transmembrane domain of MAN1. If this is accurate, please state explicitly on line 336. Would also be helpful to briefly summarize the molecular interaction between CTDNEP1 and NEP1R1 in the text.

We thank the reviewer for this suggestion. We have changed the text to specify the transmembrane segments involved in the interaction between MAN1 and NEP1R1. We believe it is unnecessary to elaborate on the details of the CTDNEP1/NEP1R1 interaction as these were not explored in the study. However, we would like to point out that that this was the subject of a recently published paper (DOI: 10.1073/pnas.2321167121), and the AlphaFold model we used is very accurate.

Lines 335-336. Figure 5H depicts CTDNEP1 as a peripheral (non-integral) membrane protein, not an integral membrane protein as strongly implied on lines 77-80. To avoid confusion, please describe their membrane-associations accurately in the introduction.

We changed the description of CTDNEP1 in the introduction.